# The occurrence of *Aerococcus urinaeequi* and non-aureus staphylococci in raw milk negatively correlates with *Escherichia coli* clinical mastitis

Dongyun Jung,[1,2,3] Soyoun Park,[1,2,3] Daryna Kurban,[2,3,4] Simon Dufour,[2,3,4] Jennifer Ronholm[1,2,3]

**ABSTRACT** *Escherichia coli* is a common environmental pathogen associated with clinical mastitis (CM) in dairy cattle. There is an interest in optimizing the udder microbiome to increase the resistance of dairy cattle to *E. coli* CM; however, the details of which members of the healthy udder microbiome may play a role in antagonizing *E. coli* are unknown. In this study, we characterized the bacterial community composition in raw milk collected from quarters of lactating Holstein dairy cows that developed *E. coli* CM during lactation, including milk from both healthy and diseased quarters (*n* = 1,172). The milk microbiome from infected quarters was compared before, during, and after CM. A combination of 16S rRNA gene amplicon and metagenomic sequencing was used generate data sets with a high level of both depth and breadth. The microbial diversity present in raw milk significantly decreased in quarters experiencing *E. coli* CM, indicating that *E. coli* displaces other members of the microbiome. However, the diversity recovered very rapidly after infection. Two genera, *Staphylococcus* and *Aerococcus*, and the family Oscillospiraceae were significantly more abundant in healthy quarters with low inflammation. Species of these genera, *Staphylococcus auricularis, Staphylococcus haemolyticus,* and *Aerocussus urinaeequi*, were identified by metagenomics. Thus, these species are of interest for optimizing the microbiome to discourage *E. coli* colonization without triggering inflammation.

**IMPORTANCE** In this study, we show that *E. coli* outcompetes and displaces several members of the udder microbiome during CM, but that microbial diversity recovers post-infection. In milk from quarters which remained healthy, the community composition was often highly dominated by *S. auricularis, S. haemolyticus, A. urinaeequi,* and *S. marcescens* without increases in somatic cell count (SCC). Community dominance by these organisms, without inflammation, could indicate that these species might have potential as prophylactic probiotics which could contribute to colonization resistance and prevent future instances of *E. coli* CM.

**KEYWORDS** mastitis, staphylococci, *E. coli*, *Aerococcus*, milk, microbiome

Bovine mastitis is an inflammation of the mammary gland of dairy cattle, mostly caused by intramammary infection (IMI). Mastitis is one of the most common and costly diseases in the dairy industry—it results in decreased quality and quantity of milk, expensive antibiotic-based treatments, and culling of chronically infected cows (1–3). Bovine mastitis causes an annual economic loss of $665 million CAD in Canada and $2 billion USD in the United States (2, 3). It is an extremely complex disease; 178 different bacterial species have been isolated from clinical mastitis (CM) samples, and scientific literature describing their impact on somatic cell count (SCC) or clinical symptoms was available for 85 of these bacterial species (4). Different etiological organisms have

Address correspondence to Jennifer Ronholm, Jennifer.Ronholm@mcgill.ca.

The authors declare no conflict of interest.

See the funding table on p. 14.

different mechanisms of transmission and result in different symptoms. Based on modes of transmission, mastitis-causing organisms can be divided into two major categories: contagious or environmental; and based on symptomology, mastitis can be clinical or subclinical (5). *Escherichia coli* is mainly associated with CM and is the most prevalent (11.0%) cause of CM in dairy cattle from Canada, United States, and Brazil, but can also be retrieved from apparently normal milking cows (i.e., subclinical mastitis) (4). It is primarily driven by environmental spread via feces, bedding, and soil on dairy farms (6). Despite improvements in hygiene practices on dairy farms, mainly aimed at controlling the spread of contagious pathogens, the number of mastitis cases caused by *E. coli* continues to increase annually (6).

*E. coli* mastitis is characterized by very acute, but generally transient infections of short duration that usually self-resolve without treatment (7–9)—although, persistent and recurrent mastitis caused by *E. coli* can occur (10). Since most cases of *E. coli* mastitis self-resolve, antibiotic treatment is not recommended (11, 12). However, severe cases can lead to septicemia and endotoxin-induced shock, and, therefore, an aggressive supportive therapy with, for instance, nonsteroidal anti-inflammatory fluids and, possibly, systemic antibiotics is important (9, 13, 14). Moreover, *E. coli* isolated from cattle with bovine mastitis are resistant to several classes of antibiotics including aminopenicillins, polypeptides, lincosamides, and macrolides (15–18). Therefore, intramammary administration of third-generation cephalosporins, specifically ceftiofur, is commonly used by producers to treat cases of severe *E. coli* CM, even though the added value of such a treatment is not demonstrated (12, 14, 19, 20). However, extended-spectrum β-lactamase (ESBL) and plasmid-mediated AmpC β-lactamase-producing *E. coli* have now been reported in cattle with mastitis, raising concerns about both human health and bovine health (21–24).

Significant effort has been put into developing vaccines targeting *E. coli* mastitis in dairy cattle, and vaccination specifically against coliform mastitis has been a part of control programs for three decades (25). However, there has been limited uptake of the available vaccines in commercial herds (26), antibody levels are known to decline over time (27), and, to maintain their efficacy, most of these vaccines have to be administered repeatedly at relatively short time-intervals, which limits their practical application.

Probiotics may be an effective way to prevent bacterial infections in the bovine mammary gland (28–30). Lactic acid bacteria (LAB) have been tested as probiotic candidates and inhibited colonization by bovine mastitis pathogens and inflammatory responses *in vitro* and *in vivo*. *Lactococcus garvieae, Lactococcus lactis, Lactobacillus brevis, Lactobacillus casei, Lactobacillus platarum,* and *Lactobacillus perolens* inhibited colonization by *E. coli, Staphylococcus aureus,* and *Streptococcus uberis* and inflammatory responses of bovine mammary epithelial cells induced by *E. coli in vitro* (28–31). *L. lactis* was also able to inhibit the growth of *S. aureus in vivo* (32). However, despite the effectiveness of *L. lactis in vivo,* it is unknown if *L. lactis* is maintained as part of the microbiome long enough to protect the host indefinitely (32). More importantly, LAB in milk can be responsible for organoleptic defects in cheese and other dairy products (33), and therefore, care should be taken when developing LAB probiotics for the dairy industry.

Optimizing the udder microbiome has been suggested as a promising way to protect dairy cattle from mastitis (34, 35), since the microbial population that inhabits mammals as part of a healthy microbiome provides a layer of protection against pathogen colonization as well as the overgrowth of opportunistic pathobionts (36). The bovine udder is home to a rich microbial community, although the composition of the bovine udder microbiome varies based on the site (37–40). The teat apex, teat canal, raw milk, and raw colostrum each have distinct microbiomes, and a healthy udder microbiome tends toward a high level of richness, while a mastitic quarter tends toward a single or a small group of bacterial species (35, 39–41). The milk microbiome from a healthy cow generally contains Firmicutes, Proteobacteria, Bacteroidetes, and Actinobacteria at the phylum level and *Staphylococcus, Ruminococcaceae,*

*Lachnospiraceae*, *Propinoibacterium*, *Stenotrophomonas*, *Corynebacterium*, *Pseudomonas*, *Streptococcus*, *Comamonas*, *Bacteriodes*, *Enterococcus*, *Lactobacillus*, and *Fusobacterium* at the genus level (42–46). Certain members of the udder bacterial community may reduce susceptibility to mastitis via one of several mechanisms. Some non-aureus staphylococci (NAS), such as *Staphylococcus chromogenes*, have been shown to antagonize growth of *Staphylococcus aureus* (47, 48). The protective effects of NAS likely occur due to the ability to produce bacteriocins (49, 50), purine analogs (51), or biofilm-disrupting signals (52). The growth of Gram-negative mastitis pathogens has not been shown to be inhibited by NAS (48). Certain NAS are also known to cause mastitis—which complicates their potential use as probiotics (53). Compared to work with *S. aureus*, relatively little work has been conducted to identify bovine commensals that may provide protection against *E. coli* CM. *Bacillus subtilis* has been shown to limit damage to mammary tissues during *E. coli* CM, although this protective effect appears to be caused by immune modulation as opposed to interbacterial antagonism (54).

In this study, we hypothesized that the raw milk microbiome in dairy cattle quarters that remained healthy would have clear and consistent differences compared to the milk from quarters that developed *E. coli* CM, and that, in identifying these differences, we may identify members of the udder microbiome that are specifically antagonistic toward *E. coli*. To test this hypothesis, we collected biweekly quarter-level milk samples from a cohort of dairy cattle that were also concurrently being monitored for *E. coli* CM. We characterized the longitudinal changes in the composition of the bacterial community in raw milk samples and performed a differential analysis comparing quarters that experienced *E. coli* CM during the study to quarters that remained mastitis-free using 16S rRNA gene-targeted amplicon sequence (TAS) analysis. To further identify members of the microbiome that negatively correlated to *E. coli* CM and further understand the *E. coli* that was causing CM, we selected certain milk samples to analyze via metagenomics. Overall, we determined that the presence of *Staphylococcus, Aerococcus*, and UCG-005 each had a negative correlation to the occurrence of *E. coli* CM.

## RESULTS

### Sample information, sequencing coverage, and overall bacterial community composition

In this study, 19.3% (135/698) of cattle developed CM during the study period. Of the 135 diagnosed CM infections, 19.25% (26/135) were caused by *E. coli*. Mastitis cases caused by *E. coli* are described in Fig. 1; Table S1. The first *E. coli* CM cases were reported across all stages of lactation: four during the transition period (1–21 days of milk [DIM]), ten in early-lactation (22–100 DIM), seven in mid-lactation (101–200 DIM), and five in late-lactation (>201 DIM). A total of 1,336 quarter-level milk samples were collected from the 26 cows diagnosed with *E. coli* CM (Table S2; Fig. 1), and DNA extraction and 16S rRNA sequencing were performed on each sample. During data processing, 209 samples were removed from the study due to low Good's coverage (< 99.0%) or a low number of sequences reads (<3,100) (Table S3). Therefore, 1,127 milk samples were included in the analysis. In addition, each individual quarter sample was plated separately on blood agar, and 41/1,127 samples produced more than two different colony morphologies. Based on guidance of the National Mastitis Council, milk samples which produce three or more colony morphologies on blood agar should not be analyzed due to possible contamination (Table S4) (55). We chose to sequence these samples despite possible contamination and analyzed data twice, both including and excluding these 41 samples. Results obtained from data including these 41 samples are reported throughout and were, for the most part, identical to results excluding these samples; however, if results differed between the two analyses, it is indicated. Sequencing efforts produced a total of 3,497,437 sequence reads passed filter (Table S3). No evidence of microbial or nucleic acid contamination was observed in any of the reagents used for DNA extraction or PCR amplification based on the sequencing results for several negative controls (Fig. S1).

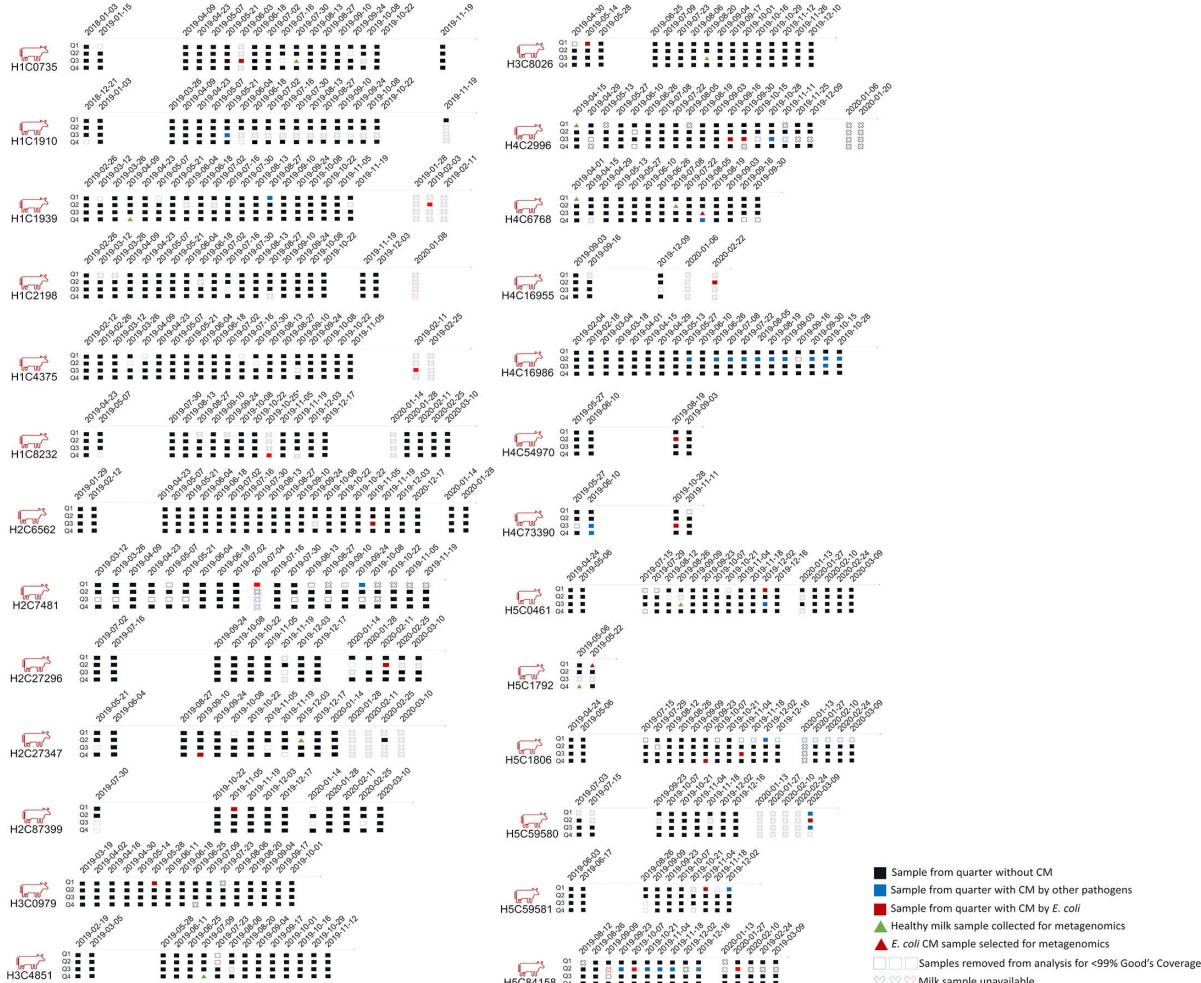

**FIG 1** Schematic representation of raw milk samples collected for this study. Raw milk samples were collected biweekly from 26 Holstein cows that developed *E. coli* CM during the study period. Each cow involved in the study is represented by an assigned cow number (ATQ), where H represents the herd number and C is the individual cow's ID number . Samples taken from cows with active *E. coli* mastitis are represented by filled red boxes, and samples taken from cows with mastitis caused by an agent other than *E. coli* are represented by filled blue boxes. Black boxes represent the samples from the quarters without CM. The samples indicated with empty boxes were excluded from the analysis due to low Good's coverage (<99.0%) or low sequence reads (<3,100). The samples indicated with X were unavailable for DNA extraction due to an insufficient volume of milk (>1.0 mL).

Milk samples from five herds shared four main phyla: Firmicutes, Proteobacteria, Bacteroidota, and Actinobacteriota (Fig. 2A and B). Firmicutes were the most dominant group across all the samples with an average relative abundance of 51.45%, followed by Proteobacteria (23.73%), Bacteroidota (11.46%), and Actinobacteriota (9.80%) (Fig. 2A). At the genus level, five most abundant OTUs represented *Staphylococcus, Aerococcus, Escherichia_Shigella,* and UCG-005 from the a family and an unclassified genus from the Enterobacteriaceae family (Fig. 2C). The Chao1 index was significantly different between herd 1 and herd 5 and herd 3 and herd 5 (TukeyHSD *post-hoc* test; *P* < 0.01) (Fig. S2). The permutational multivariate analysis of variance (PERMANOVA) analysis of the Bray–Curtis dissimilarity showed that beta diversity was significantly different between herds (PERMANOVA; *P* < 0.01) (Fig. S2).

## Microbial changes in milk before, during, and after *E. coli* CM

The diversity of the milk microbiome was compared over time between quarters which developed *E. coli* CM and control quarters that did not develop *E. coli* CM during the study period. Alpha diversity was not significantly different between these categories

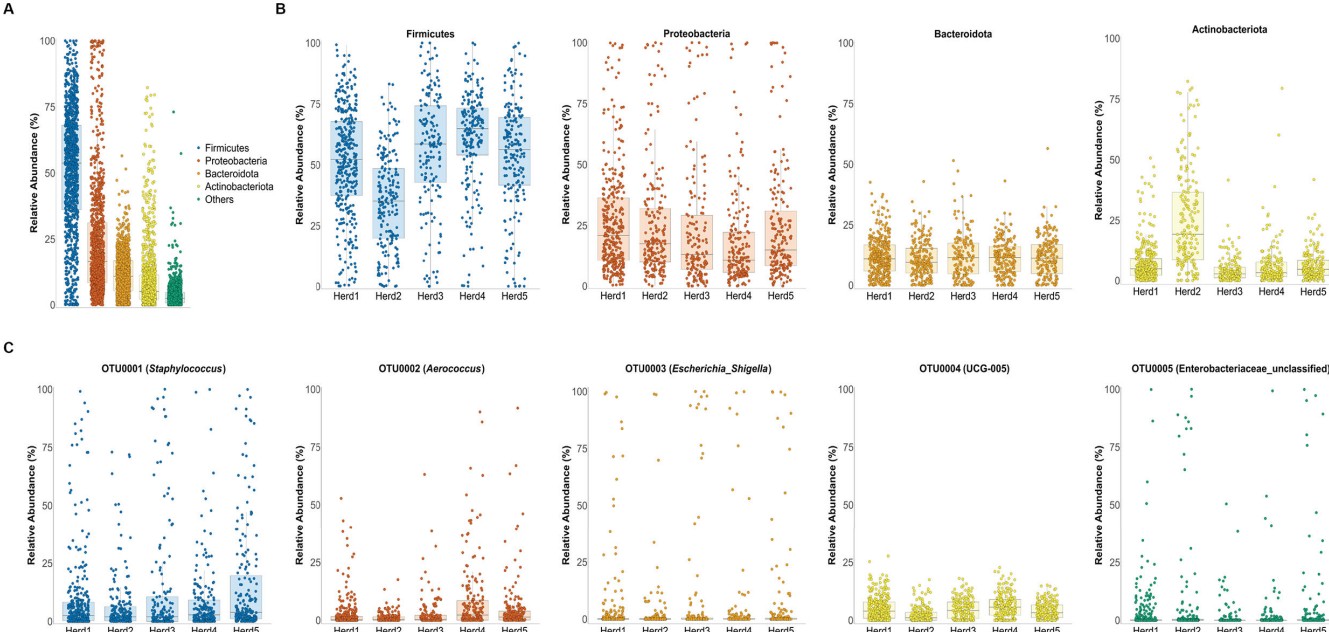

**FIG 2** Relative abundance of phyla and OTUs in raw milk samples. (A) The relative abundance of each phylum in every sample is represented by a single data point, where Firmicutes, Proteobacteria, Bacteroidota, and Actinobacteriota were the most common phyla present in milk. (B) The relative abundance of each of the four most common phyla is shown for each of the five herds included in the study. (C) The relative abundance of each of the five most common OTUs is shown for each of the five herds included in the study.

of quarters if the milk was collected 2 weeks before or after *E. coli* CM; however, alpha diversity was significantly different on the day of *E. coli* CM diagnosis (Shannon *P* < 0.01; Mann–Whitney statistic 84.5, Simpson *P* < 0.01; Mann–Whitney statistic 86.5, Chao1 *P* < 0.01; Mann–Whitney statistic 196.5) (Fig. 3A; Table 1). However, when samples were analyzed without the 41 samples that produced more than two colonies on blood agar, Chao1 was not significantly different on the day of *E. coli* CM (Chao1 *P* = 0.0104; Mann–Whitney statistic 236.0). The difference in alpha diversity on the day of CM was driven primarily by the dominance of the *Escherichia-Shigella* OTU (Fig. S3; Table 1). Differences in beta diversity were also only significant on the day of CM based on PERMANOVA analysis of the Bray–Curtis dissimilarities (PERMANOVA *P* < 0.01, F = 15.169) (Fig. 3B; Table 2). The relative abundances of OTU0001 (*Staphylococcus*), OTU0002 (*Aerococcus*), and OTU0004 (UCG_005) were significantly higher in milk taken from healthy quarters compared to milk taken from *E. coli* CM quarters on the day *E. coli* CM was diagnosed (Mann–Whitney U test adjusted by Benjamini & Hochberg (BH); *P* < 0.01) (Fig. 3C; Fig. S3). These correlations were also identified using linear discriminant analysis (LDA) where *Escherichia_Shigella* was highly associated with milk from quarters with *E. coli* CM (LDA = 5.49), and *Staphylococcus*, *Aerococcus*, and UCG-005 were highly associated with milk from healthy quarters (LDA < −4.1) (Kruskal–Wallis rank-sum test; *P* < 0.01) (Fig. 3D). Other OTUs were also associated with milk from healthy quarters on the day of *E. coli* CM, including *Ruminococcaceae*, *Bifidobacterium*, *Bacteroides*, *Atopostipes*, and *Jeotgalicoccus* (LDA > 2.0 or < −2.0; Kruskal–Wallis rank-sum test *P* < 0.01) (Table 3).

## Microbial network analysis of the bacterial community

To determine potentially biologically relevant interactions that occur between each of the OTUs found in milk, the 187 OTUs that had more than 10% of prevalence across 1,127 samples were further investigated (Fig. S4; Table S5). No significant associations were identified between any OTUs (−0.165 < Spearman's ρ < 0.666).

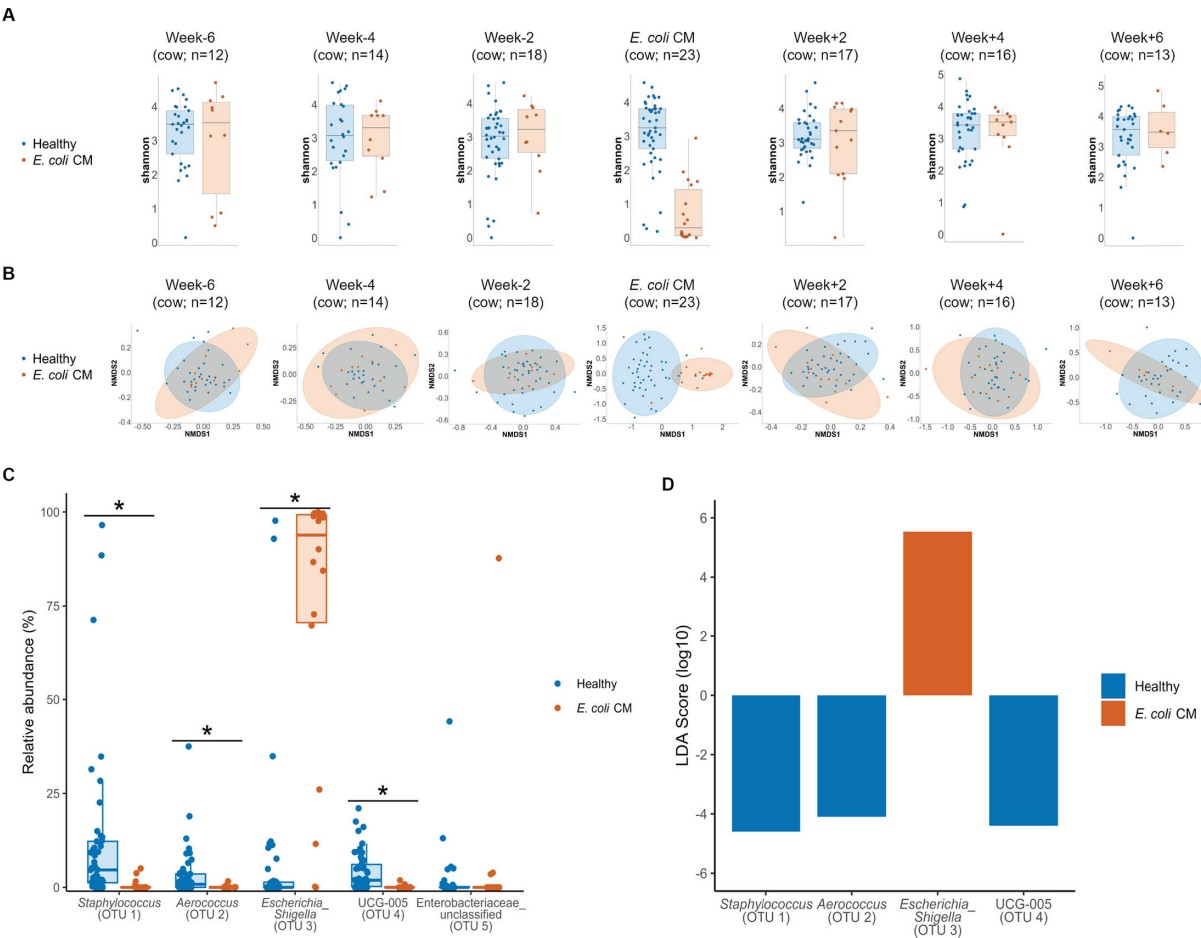

FIG 3 Longitudinal changes in alpha and beta diversity in raw milk samples. The alpha (A) and beta (B) diversity of the microbial community are displayed, based on estimates at the OTU level, before, during, and after *E. coli* CM using the Shannon index to plot alpha diversity and NMDS to plot beta diversity. The $R^2$ values for each NMDS plot are 0.03078, 0.02944, 0.01915, 0.16978, 0.02018, 0.02587, and 0.02587 from left to right. Differential abundance analysis is plotted using percent relative abundance (C) and LDA score (D) for the five most common OTUs in the study to demonstrate differences between milk samples taken from cows with and without *E. coli* CM—all OTUs in the study were considered in the differential abundance analysis.

## Correlations between SCC and bacterial abundance and diversity

The correlation between the log SCC and the Shannon index in milk samples from quarters that did not have *E. coli* CM (*n* = 844) was evaluated to determine if there was a clear relationship between bacterial alpha-diversity and inflammation in the absence of CM. The Shannon index of the samples had a weak correlation with the log SCC (Spearman's ρ = 0.009) (Fig. 4A). The correlations between the log SCC and each of the five most abundant OTUs in healthy samples were also evaluated, but all OTUs showed a weak correlation (0 < Spearman's ρ < 0.1) (Fig. 4B)

## Metagenomic sequencing analysis of milk from healthy quarters

Select samples (*n* = 11) with a high relative abundance (> 40%) of each of the two most common OTUs and a low SCC (< 50,000 cells/mL; log SCC <4.7) were selected for metagenomic sequencing to identify bacteria, at the species level, that may outcompete *E. coli* in this environment but do not result in inflammation (Fig. 4B). A total of 476,966,298 raw sequence reads were produced (Table S6). Filtering was performed to remove any sequence reads that corresponded to bovine DNA. After bovine DNA was removed, 36,259,190 sequence reads remained (Table S7). All samples underwent MAG analysis; however, only sample, 30000729, yielded high-quality MAGs. Three MAGs:

**TABLE 1** Statistical analysis of alpha diversity (Mann–Whitney U) in the milk microbiome before, during, and after the first *E. coli* CM case

| Alpha diversity index | Week | *W*-value | *P*-value |
|---|---|---|---|
| Chao1 | Week 6 | 167 | 0.8478 |
| | Week 4 | 139 | 0.6509 |
| | Week 2 | 309.5 | 0.03258 |
| | *E. coli* CM | 184 | 0.00008784 |
| | Week 2 | 316 | 0.09714 |
| | Week 4 | 185 | 0.8569 |
| | Week 6 | 97.5 | 0.8796 |
| Shannon | Week 6 | 166 | 0.871 |
| | Week 4 | 137 | 0.6066 |
| | Week 2 | 243 | 0.5319 |
| | *E. coli* CM | 68 | 0.00000005654 |
| | Week 2 | 251 | 0.825 |
| | Week 4 | 194 | 0.9795 |
| | Week 6 | 112 | 0.719 |
| Simpson | Week 6 | 163 | 0.9411 |
| | Week 4 | 134 | 0.5428 |
| | Week 2 | 243 | 0.5319 |
| | *E. coli* CM | 77 | 0.0000001082 |
| | Week 2 | 206 | 0.4521 |
| | Week 4 | 206 | 0.7378 |
| | Week 6 | 113 | 0.6908 |

*Aerococcus urinaeequi* (genome size: 1,859,582 bp; CDS: 1,683; completeness = 97.80%, contamination = 1.10%), *Staphylococcus auricularis* (genome size: 2,006,007 bp, CDS: 1,889; completeness = 98.34%, contamination = 0.0%), and *S. haemolyticus* (genome size: 2,623,558 bp; coding sequences (CDS): 2,603; completeness = 98.48%, contamination = 1.99%), were assembled from sample 30000729. Sample 30000729 was taken from an exceptionally healthy quarter that did not develop CM from any pathogen during the study period and had an SCC of 5,000 on the day the sample was collected.

Overall, taxonomic profiling was performed on these 11 metagenomes, which identified the presence of 12 species: *Acinetobacter gandensis, A. lwoffii, A. towneri, A. urinaeequi, C. bovis, C. stationis, Glutamicibacter arilaitensis, Luteimonas* sp. J29, *P. fluorescens, Serratia marcescens, S. auricularis,* and *S. haemolyticus* (Fig. 5; Table S8).

Milk sample 50000827 was selected for metagenomic sequencing since it was identified as having a high abundance of the *Escherichia-Shigella* OTU (90.42%), but a low SCC/mL of milk of only 44,000. The combination of a high abundance of *E. coli* in milk combined with a low SCC led us to question if certain strains of *E. coli* could act as commensal organisms in the bovine udder and not result in inflammation—something that has not been reported. There were several milk samples in our data set that had a high relative abundance of *Escherichia-Shigella* but low SSC (Fig. 4), but we chose to perform metagenomic sequencing 50000827 because the extremely high relative

**TABLE 2** Statistical analysis of beta diversity (PERMANOVA) in milk microbiome before, during, and after the first *E. coli* CM case

| Index | Week | F-value | *P*-value |
|---|---|---|---|
| Bray–Curtis | Week 6 | 1.1948 | 0.1149 |
| | Week 4 | 0.02629 | 0.4386 |
| | Week 2 | 0.97508 | 0.5165 |
| | *E. coli* CM | 12.679 | 0.000999 |
| | Week 2 | 1.0098 | 0.4326 |
| | Week 4 | 1.1942 | 0.1329 |
| | Week 6 | 0.93665 | 0.5415 |

**TABLE 3** Statistical analysis of differentially abundant bacterial taxa before, during, and after the first *E. coli* CM case

| Test | Week | OTU | LDA | *P*-value | Genus (group with higher abundance) |
|---|---|---|---|---|---|
| LEfSe | Week 6 | Otu0027 | −4.03405 | 0.00683636 | *Jeotgalicoccus* (healthy) |
| | Week 2 | Otu0028 | 2.41828 | 0.00451753 | *Facklamia* (*E. coli* CM) |
| | | Otu0078 | 3.49283 | 0.00918033 | *Thermicanus* (*E. coli* CM) |
| | | Otu0087 | 3.33617 | 0.00134171 | Unclassified Myxococcaceae (*E. coli* CM) |
| | | Otu0099 | 3.05235 | 0.0000173 | Lachnospiraceae_NK3A20_group (*E. coli* CM) |
| | *E. coli* CM | Otu0001 | −4.67254 | 0.000017 | *Staphylococcus* (healthy) |
| | | Otu0002 | −4.15486 | 0.0000648 | *Aerococcus* (healthy) |
| | | Otu0003 | 5.49274 | 0.0000000961 | *Escherichia_Shigella* (*E. coli* CM) |
| | | Otu0004 | −4.33813 | 0.00000591 | UCG-005 (healthy) |
| | | Otu0008 | −3.99625 | 0.00953077 | UCG-005 (healthy) |
| | | Otu0010 | −4.08372 | 0.00794878 | Ruminococcaceae (healthy) |
| | | Otu0011 | −4.04736 | 0.00255481 | *Bifidobacterium* (healthy) |
| | | Otu0016 | −3.8713 | 0.00149518 | *Bacteroides* (healthy) |
| | | Otu0017 | −3.85045 | 0.00245468 | *Atopostipes* (healthy) |
| | | Otu0027 | −3.79745 | 0.00661443 | *Jeotgalicoccus* (healthy) |
| | Week 2 | Otu0020 | 3.59003 | 0.00751313 | UCG-005 (*E. coli* CM) |
| | | Otu0030 | 3.796 | 0.00391903 | *Ruminobacter* (*E. coli* CM) |
| | | Otu0086 | 3.50007 | 0.00195887 | *Bacteroides* (*E. coli* CM) |
| | Week 6 | Otu0183 | 3.84213 | 0.00667029 | *Coxiella* (*E. coli* CM) |
| | | Otu0268 | 3.05589 | 0.00243179 | *Alishewanella* (*E. coli* CM) |

abundance of the *Escherichia-Shigella* OTU in the sample would allow for the best chance at assembling an MAG. Metagenomic sequencing revealed a single MAG of *S. marcescens* (genome size: 2,840,286 bp, CDS: 2,672; completeness = 71.29%, contamination = 1.84%) from this sample, and no *E. coli* was identified from the metagenomic sequences. This led us to question if there was enough similarity between the V4 region of *Escherichia-Shigella* and *S. marcescens*, where *S. marcescens* 16S rRNA sequences would be assigned to this OTU. To examine the similarity of the V4 region in the 16S rRNA gene between *E. coli* and *S. marcescens*, we aligned the sequences of the V4 region (Fig. S5). The alignment showed 98.8% identity, which was above the threshold to assign OTUs, indicating that *S. marcescens* sequences are also included in the *Escherichia-Shigella* OTU. Therefore, no evidence of a commensal mammary *E. coli* was identified in this investigation.

Annotation of the MAGs identified several genes that could contribute to *E. coli* antagonism including the D-lactate dehydrogenase gene, *ldhA*, in the MAGs of *S. haemolyticus* and *S. marcescens*, and the L-lactate dehydrogenase gene, *ldh*, in the MAGs of *S. auricularis*, *S. haemolyticus*, and *A. urinaeequi*. The genes of class IIa bacteriocins listeriocin and pediocin were identified in the MAG of *A. urinaeequi*. The *S. marcescens* MAG contained the genes of ten core components (TssA, TssE, TssF, TssG, TssH, TssJ, TssK, TssL, and TssM) and two accessory components (Hcp and VgrG) of a type VI secretion system (T6SS).

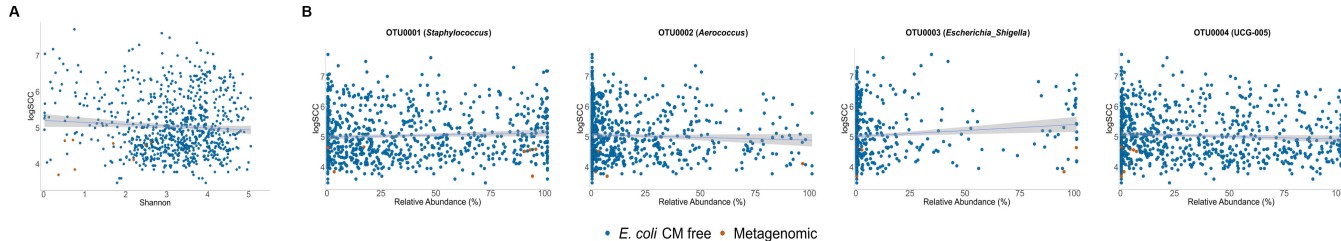

**FIG 4** Correlations between inflammation and the raw milk microbiome composition. The Shannon index (A) or the relative abundance of the four most common OTUs (B) plotted with logSCC to visualize the relationship between bacterial diversity or specific taxa and inflammation for each sample.

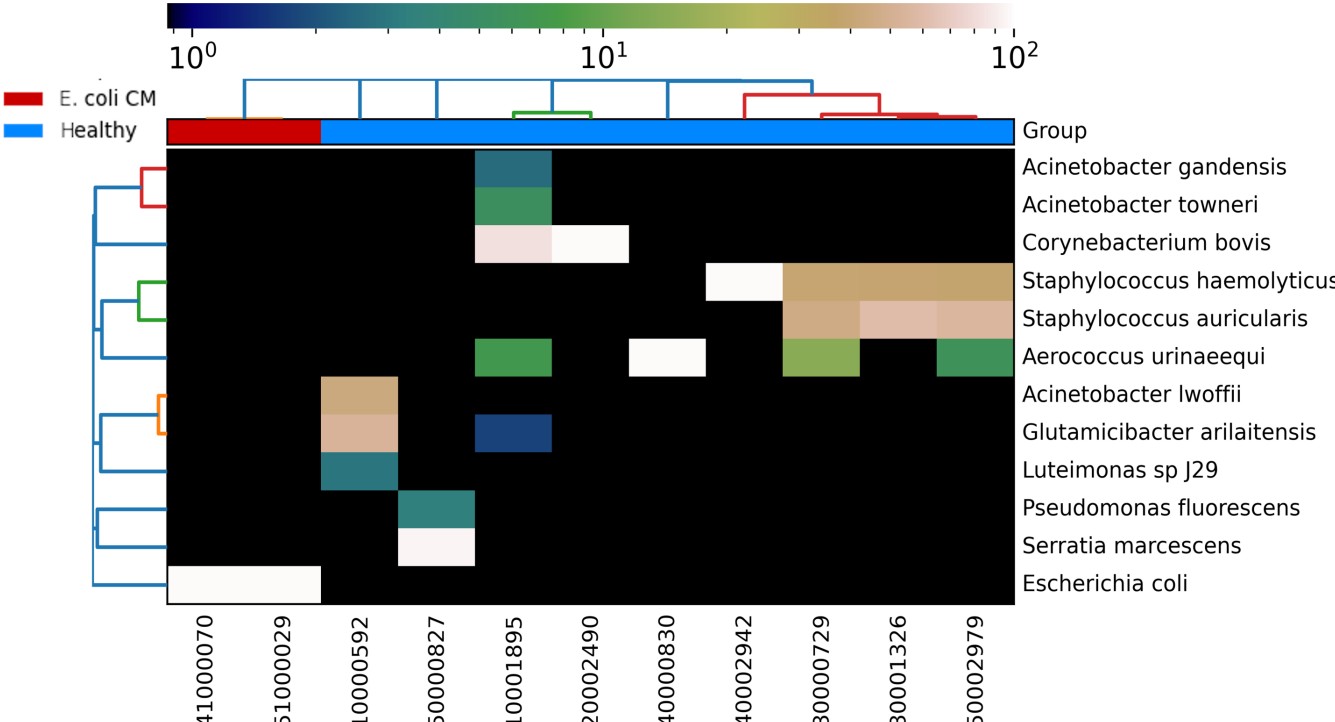

**FIG 5** Metagenomic sequencing analysis of the microbial composition of healthy and *E. coli* CM samples at species level. The relative abundance of taxonomic composition of the metagenomic sequencing is shown using a heat map where sample IDs are displayed on the *x*-axis and bacterial species with >0.1% of the relative abundance of sequence reads are shown on the right *y*-axis. The phylogenetic relatedness of the bacterial species identified is shown on the left *y*-axis, and the phylogenetic relatedness of the metagenomic sequences is shown at the top of the figure.

## Metagenomic analysis of milk from *E. coli* CM quarters

Two milk samples with *E. coli* CM (41000070 and 51000029) were selected for metagenomic sequencing to provide additional insights into the pathogen and the composition of the bacterial community during *E. coli* CM. A total of 113,859,360 raw sequence reads were produced from these two samples (Table S6). Filtering was performed to remove any sequence reads that corresponded to bovine DNA. After bovine DNA was removed, 172,994 sequence reads remained (Table S7). All sequence reads were identified as *E. coli*.

## DISCUSSION

We investigated the bacterial composition in raw milk taken before, during, and after *E. coli* CM from both healthy and CM quarters of 26 dairy cows that developed natural *E. coli* CM during our study period. We conducted 16S rRNA amplicon sequencing to track the changes of the microbiome in the milk samples from the quarters that developed *E. coli* CM and the quarters that remained healthy for the study. Metagenomic analysis was conducted on the samples taken from healthy quarters to profile potential commensal bacteria at the species level that may outcompete or be actively antagonistic toward *E. coli*.

The microbiome in raw milk samples was dominated by four phyla: Firmicutes, Proteobacteria, Bacteroidota, and Actinobacteriota. We observed the variations in the abundance of these taxa across herds and cows, except Bacterioidota, which remained at a constant low level. This observation is similar to that of previous studies that demonstrated microbiome variation from the farm-to-farm level to the cow level (56–58). Actinobacteriota was significantly more abundant in herd 2 than in the other herds. *Glutamicibacter*, a genus from Actinobacteriota, accounted for most of the Actinobacteria in herd 2. The number of *E. coli* CM cases in herd 2 was not significantly different from that in other herds. Some genera of Actinobacteriota phyla, such as *Streptomyces* and

*Bifidobacterium*, have been reported to have antimicrobial activities against *E. coli* (59, 60); however, each was present at very low relative abundances in our data set.

We investigated changes in bacterial community composition in the samples taken from all quarters of 26 cows before, during, and after the first *E. coli* CM and observed that milk samples taken during active *E. coli* CM mastitis had significantly decreased alpha diversity, but this quickly rebounded after the infection. This indicates that pathogen clearance and the reestablishment of the udder microbiome happen quickly after infections, which has been observed previously (43, 61). In earlier work, experimental infection with *E. coli* led to a peak in *E. coli* counts 16–24 hours post-infection, and bacterial clearance without treatment 7 days after a challenge, although, about 50% of cows intermittently shed *E. coli* asymptomatically until the end of the study period (61). A decrease in bacterial diversity in milk samples prior to CM has been observed for other mastitis pathogens and may even be predictive for a future infection in the case of *S. aureus*, but observations from our study indicate that diversity alone is not predictive of which quarters will develop *E. coli* CM (37, 56).

The differential abundance analysis of various bacterial taxa between healthy samples and samples taken from quarters with *E. coli* CM identified a significant increase in the abundance of *Escherichia-Shigella* in mastitic milk and a significant increase in the abundance of *Staphylococcus*, *Aerococcus*, and UCG-005 in healthy samples. *Staphylococcus*, *Aerococcus*, and UCG-005 were also highly dominant in some healthy samples that had less than <50,000 cells/mL of SCC (Fig. 4B). The dominance of these genera in healthy quarters, without triggering inflammation, may imply that these members may be able to outcompete pathogenic species in this environment and should be investigated further as potential probiotics. A major limitation of the current study is that co-culture tests to determine if antagonistic activity does exist between *Staphylococcus* and *Aerococcus* isolates and *E. coli* in the context of the bovine udder were not performed. However, this will be the focus of much future work. The lack of a significant association between the SCC and Shannon index and the high relative abundance (>40%) of some OTUs in the healthy samples that had less than 50,000 cells/mL of the SCC also implies a small impact of dysbiosis on the susceptibility to the disease.

To find the specific species of *Staphylococcus*, *Aerococcus*, and UCG-005 that could dominate a microbial community without causing inflammation, we performed metagenomic sequencing of samples with a high abundance of *Staphylococcus* or *Aerococcus* but a low SCC. This identified *S. auricularis*, *S. haemolyticus*, and *A. urinaeequi* at the species level. *S. auricularis* and *S. haemolyticus* are NAS and have previously been reported to cause subclinical mastitis and CM (53, 62–68). However, a previous study reported that *S. auricularis* and *S. haemolyticus* could be present on the teat apices of dairy cows without causing IMI (62); other recent studies found that NAS makes up a significant portion of the healthy udder microbiome and that they are predominant in the raw milk samples (SCC < 200,000 cells/mL) from the healthy quarters of Holstein dairy cows (53, 56, 69). Furthermore, NAS species including *S. capitis*, *S. chromogenes*, *S. epidermidis*, *S. pasteuri*, *S. saprophyticus*, *S. sciuri*, *S. simulans*, *S. warneri*, and *S. xylosus* can inhibit the growth of *S. aureus* associated with clinical or subclinical mastitis by bacteriocins and the purine analog 6-thioguanine (49–51). However, the inhibition of growth of *E. coli* by NAS has not been previously reported prior to this study.

The MAGs of *S. auricularis*, *S. haemolyticus*, and *A. urinaeequi* were constructed from a healthy milk sample, and each genome encodes D- and L-lactate dehydrogenase, which catalyzes the production of lactic acid from pyruvate. Several studies have reported that lactic acid can inhibit the growth of *E. coli* O157:H7 *in vitro* (70–72). Considering the previous studies that tested LAB as probiotic candidates to prevent IMI and the high prevalence of L-lactate dehydrogenase across the healthy samples (28–32, 73), *S. auricularis*, *S. haemolyticus*, and *A. urinaeequi* could be capable of prevention while being maintained long enough as commensal groups.

*A. urinaeequi* has rarely been identified as a bovine mastitis pathogen, while *Aerococcus viridans*, a close phylogenetically related species (74), is known as a subclinical

mastitis pathogen (75, 76). *A. urinaeequi* has demonstrated antimicrobial activity against *Klebsiella pneumoniae, Salmonella enterica, Vibrio alginolyticus,* and *S. aureus* (56, 77). A newly identified class II bacteriocin from *A. urinaeequi* has shown antagonistic activity against *K. pneumoniae, S. enterica,* and *V. alginolyticus*, but the activity against *E. coli* by the bacteriocin has not been identified yet (77).

An MAG identified as *S. marcescens* was constructed from a healthy milk sample (5000827); the annotation of the MAG identified genes of T6SS components comprising the genes of ten core components (TssA, TssE, TssF, TssG, TssH, TssJ, TssK, TssL, and TssM) and two accessory components (VgrG and Hcp). Previous studies found that *S. marcescens* could have T6SS and inhibit the growth of *E. coli in vitro* (78, 79). *S. marcescens* is an etiological agent of bovine clinical mastitis (80). However, the milk sample we identified as containing a high abundance of *S. marcescens* had an SCC of 44,000 cells/mL, indicating a relatively healthy quarter. These findings could indicate that this strain of *S. marcescens* does not illicit significant inflammation.

In this study, we characterized changes of the microbial community before, during, and after *E. coli* CM in the bovine udder and identified that commensal bacteria in the samples from healthy quarters correlate negatively to the presence of *E. coli*. An unbalanced microbiome, driven by the overgrowth of certain commensals, is not necessarily associated with intramammary inflammation and bovine mastitis. In our study, we identified four such species, and the assembled MAGs of these species each encoded mechanisms that can potentially antagonize *E. coli*. The modulation of the microbiome with these species could be a potential way of preventing future *E. coli* CM without inducing inflammation. Further studies need to focus on the isolation of these species from the healthy cows from a wide range of geographical origins and their antagonistic effect on *E. coli in vitro* and *in vivo* as potential probiotics.

## MATERIALS AND METHODS

### Sample collection

The study was affiliated with Park et al., as part of a larger project that included 698 Holstein dairy cows from five herds in the Quebec province, in proximity to the Faculty of Veterinary Medicine at Université de Montréal in Saint-Hyacinthe (56). Raw milk samples were collected aseptically from each quarter every 2 weeks during the lactation period between December 2018 and March 2020. Briefly, the farm's teat disinfectant was applied, and after a contact time of 20 seconds, the teats were dried using a paper towel, and each teat end was then scrubbed with individual alcohol swabs. Approximately 60 mL of milk was then discarded, and 50 mL was collected (i.e., cisternal milk). Milk samples were placed on ice and transported, on the same day, to the Faculty of Veterinary Medicine laboratory where they were stored at 4°C prior to being aliquoted for the different parts of the study. After being aliquoted, samples were frozen. The SCC was measured for each milk sample from quarters without any clinical signs of mastitis. In parallel to these farm visits, CM was identified by the producers by visual inspection of the milk and udders, and a milk sample was collected from the affected quarter by the producers at diagnosis, following the protocol previously described. These latter samples were immediately frozen on the farm at −20°C, collected by the research team during the next farm visit, and processed in the laboratory as previously described.

The National Mastitis Council's guidance on raw milk sampling advises that if a milk sample is plated on blood agar and after 24 or 48 hours (when the sample yielded no growth after 24 hours of incubation) of incubation >2 different colony morphologies are observed, the sample is considered contaminated (55). Therefore, after collection, 10 µL of each raw milk sample was spread on 5% sheep blood agar and incubated for 24 or 48 hours at 35°C. The number of phenotypically different colonies that were grown for each milk sample was counted and recorded (Table S4) (81). If a sample produced ≥2 colony morphologies, colony identification was not performed on these samples, and they were classified as contaminated (Fig. S6). Colony identification was

not performed on samples which produced >2 colony morphologies. However, DNA for 16S rRNA TAS was still extracted from these samples and the data set was analyzed with and without putatively contaminated samples. For samples which produced two or less colony morphologies on blood agar, a representative colony for each colony type was identified using matrix-assisted laser desorption/ionization time-of-flight (MALDI-TOF) mass spectrometry (Table S4). All the collected milk samples were then stored between −10°C and −20°C until bacterial DNA extraction.

Samples were classified as being from a healthy quarter if they were collected from a quarter with no clinical symptoms of mastitis and a SCC of <200,000 cells/mL of milk. If a quarter milk sample came from a quarter which had signs of CM, the sample was classified as "clinical mastitis." If a quarter milk sample had an SCC of >200,000 cells/mL of milk and yielded a positive culture for *E. coli,* the sample was classified as having subclinical mastitis caused by *E. coli* (Fig. S6).

## Bacterial DNA extraction

All samples were transported to McGill University for bacterial DNA extraction. Bacterial DNA was extracted from each milk sample taken from an animal that developed *E. coli* CM during the study period. The samples were taken from 6 weeks before to 6 weeks after the first *E. coli* CM (Fig. 1). The DNeasy PowerFood Microbial Kit (QIAGEN, Germany) was used following the manufacturer's instructions, with minor exceptions. To prepare milk samples for DNA extraction, frozen milk was fully thawed on ice and inverted several times to mix, and then a 1-mL aliquot was placed in an Eppendorf microcentrifuge tube centrifuged at 16,000 × g for 10 minutes. The supernatant was discarded, and the DNA extraction protocol was carried out using the pellet. Separate negative and positive extraction controls were included in the analysis for each reagent kit used. For a negative control, the extraction procedure was carried out using DNA-/RNA-free water in place of a milk pellet. The positive control consisted of extracting DNA, using the same kit, from a rumen sample from a generous donor. To extract a higher quantity of DNA for metagenomic sequencing, a 6.0-mL aliquot of milk was used. The sample was prepared by centrifuging 1 mL of milk at 16,000 x g for 10 minutes, discarding the supernatant, then adding another 1 mL of fresh milk to the pellet, and repeating this procedure five times. The DNA was extracted from the larger pellet using the same protocols as described above. The concentration and purity of extracted DNA from all samples and the controls were evaluated using the Invitrogen Quant-iT dsDNA Assay Kit (Thermo Fisher Scientific, USA) and a NanoDrop 2000 (Thermo Scientific, USA), and samples with poor quality or quantity were re-extracted.

## 16S rRNA gene sequencing

The V4 hypervariable region of the bacterial 16S rRNA gene was amplified from the extracted bacterial DNA from milk samples (*n* = 1,336) using PCR with the F515 and R806 primer pair and sequenced using Illumina MiSeq platform using paired-end sequencing (2 × 251 bp) (Illumina Inc., San Diego, CA, USA) (82). A negative kit control (*n* = 15), which included extracting DNA from DNA-/RNA-free water using each of the reagents available in the DNA extraction kit, was prepared for each DNA extraction kit used. An independent negative PCR control (*n* = 15), which consisted of an attempt to amplify DNA-/RNA-free water, was included for each 96-well PCR performed as part of this study. Both kit controls and PCR controls were also sequenced. The HotStartTaq Plus Master Mix Kit (QIAGEN, Germany) was used for PCR, and the amplification cycle included initial denaturation at 95°C for 5 minutes, 35 cycles of denaturation at 95°C for 30 seconds, annealing at 50°C for 30 seconds, extension at 72°C for 1 minute, and final extension at 72°C for 10 minutes. The amplicon libraries were purified using Agencourt AMPure XP (Beckman Coulter, Brea, CA, USA) as per the manufacturer's instructions and quantified using the Invitrogen Quant-iT dsDNA Assay Kit. The libraries were then normalized and pooled (>1 nM) followed by sequencing using the MiSeq reagent kit V2 for 502 cycles (2 × 251 bp) and the MiSeq benchtop sequencer.

## 16S rRNA gene sequencing data analysis

FASTQ files were generated from the Illumina MiSeq sequencer and analyzed using Mothur (v. 1.42.3) (83). Mothur aligned the read pairs to make contigs with the length 253 bp, and the similar contigs were clustered and aligned by the SILVA reference (v. 138) (84). Based on alignment to the SILVA reference, the *E. coli* causing CM were classified as *Escherichia-Shigella* at the genus level. Then, chimeric sequences were removed using UCHIME, and the filtered contigs were clustered and assigned to operational taxonomic units (OTUs) at 97% of identity as a threshold (85). The average OTUs were obtained by rarefying the sequences 1,000 times repeatedly to the minimum number of sequences ($n$ = 3,100) using the Vegan R package (vegan::rarefy.perm) (v. 2.6–2) (86, 87). Following the rarefaction, Good's coverage of the sequences was calculated on R (v. 4.1.2), and all samples with higher than 99.0% Good's coverage were retained for analysis (88).

## Diversity and relative abundance of abundant taxonomic groups analysis

Differences in alpha diversity (Chao1, Shannon, and Simpson) and beta diversity (Bray–Curtis) were compared between milk samples collected from healthy quarters and quarters that developed *E. coli* CM using the vegan R package. The Mann–Whitney U test and PERMANOVA test were conducted to identify significant differences in alpha and beta diversities between the sample groups. Non-metric multidimensional scaling (NMDS) ordination was used to plot the beta diversity using the Vegan R package. The Mann–Whitney U test and linear discriminant analysis effect size (LEfSe) were conducted to determine the differential abundance of OTUs between the two types of samples. The OTUs with LDA scores higher than 2.0 from LEfSe and BH-adjusted $P$-value lower than 0.01 were considered to be significant (89). Spearman's correlation and standard linear regression were used to characterize the relationships between log (SCC) and Shannon index and between log (SCC) and the abundant OTUs on R.

## Microbial network analysis

The microbial network from all milk samples was created using the rarefied OTU table in R. Each network was created by calculation of co-occurrence using the Spearman's correlation between the OTUs and corroborated with two OTU-generalized linear models (GLM)—one that included only environmental independent variables and another that included independent variables and the relative abundance of each other's OTUs (56, 90). Cows and quarters as environmental independent variables and quasi-Poisson distribution on each OTU–OTU combination were used for GLM analysis. The OTU–OTU correlations that had a $P$-value > 0.01, discrepant results between the Spearman and GLM analyses, and a Spearman's $\rho \leq 0.2$ and $\geq -0.2$ were excluded (90, 91). The standardized $\beta$ coefficient values were used to visualize the networks on Cytoscape (v. 3.8.2) (92).

## Metagenomic sequencing

The samples selected for metagenomic sequencing included two milk samples taken from *E. coli* CM quarters on the day of diagnosis and 11 milk samples from healthy quarters each from different cows. Samples used for metagenomic sequences are indicated in Fig. 1. Healthy milk samples were chosen on the basis that they had more than 40% relative abundance of one of the five most abundant OTUs and low SCC (<50,000 cells/mL). This selection criteria were chosen in an attempt to a high enough concentration of sequence reads aligned to the OTU to assemble a metagenome-assembled genome (MAG). After DNA extraction, the Nextera XT DNA Flex Library Preparation Kit (Illumina Inc., San Diego, CA, USA) and Nextera XT Index Kit were used to prepare metagenomic libraries following the manufacturer's instructions. Then, paired-end sequencing (2 × 150 bp) was performed using the Illumina NovaSeq 6000 sequencer by Genome Quebec (Montreal, QC, Canada).

## Metagenomic analysis

BioBakery tools were used for the analysis of the metagenome sequences (93). Raw FASTQ files were processed using KneadData (v. 0.10.0), which filtered out Illumina Nextera adapters and the host genome (https://github.com/biobakery/kneaddata). *Bos taurus* 3.1 (UMD 3.1, https://bovinegenome.elsiklab.missouri.edu/downloads/UMD_3.1) was a reference host genome. The threshold of the quality score was 30, and highly duplicated sequences (>40%) were de-duplicated using the clumpify.sh function by BBMap (v.38.86) (94). The metagenomic data were analyzed using both genome binning and assembly-free approaches.

For genome binning, the filtered reads were assembled by metaSPAdes (v. 3.15.4) followed by taxonomic binning by metaBAT2 (v. 2.14) and MaxBIN2 (v. 2.2.7) (95–97). The bins were refined using MAGpurify (v. 2.1.2) and evaluated by CheckM (v. 1.0.18), which filtered out the bins with less than 70% of completeness and more than 2.0% of contamination (98, 99). The species of MAGs from the successful bins were identified using the Genome Taxonomy Database Toolkit (GTDB-Tk) (v. 2.2.3) with the reference database, R207_v2, and the annotation was done using Prokka (v. 1.14.5), BAGEL4, AntiSMASH, and RASTtk (v. 2.0) on BV-BRC (v. 3.30.19) (100–104).

For assembly-free analysis, the filtered reads after KneadData were processed by MetaPhlAn 4 (v. 4.0.3) to identify the taxonomic groups at the species level using the species-level genome bin (SGB) database (mpa_vJan21_CHOCOPhlAnSGB_202103) with the default parameters (105–107). Heatmaps of the taxonomic profiles were plotted using hclust2 with the default parameters, except for Bray–Curtis as the distance function for samples (https://github.com/segatalab/hclust2).

## ACKNOWLEDGMENTS

The sample collection was financially supported by a funding from the Dairy Research Cluster 3 (Dairy Farmers of Canada [Ottawa, ON, Canada] and Agriculture and Agri-Food Canada [Ottawa, ON, Canada]) under the Canadian Agricultural Partnership AgriScience Program and by the Mastitis Network (Saint-Hyacinthe, QC, Canada).

## AUTHOR AFFILIATIONS

[1]Faculty of Agricultural and Environmental Sciences, Macdonald Campus, McGill University, Quebec, Canada
[2]Mastitis Network, Saint-Hyacinthe, Québec, Canada
[3]Regroupement FRQNT Op+Lait, Saint-Hyacinthe, Québec, Canada
[4]Faculté de Médecine Vétérinaire, Université de Montréal, Saint-Hyacinthe, Québec, Canada

## AUTHOR ORCIDs

Dongyun Jung http://orcid.org/0000-0003-4932-4981
Simon Dufour http://orcid.org/0000-0001-6418-0424
Jennifer Ronholm http://orcid.org/0000-0001-7902-3368

## FUNDING

| Funder | Grant(s) | Author(s) |
| --- | --- | --- |
| Dairy Farmers of Canada (DFC) | Dairy Research Cluster 3 | Simon Dufour |
| | | Jennifer Ronholm |

## AUTHOR CONTRIBUTIONS

Dongyun Jung, Conceptualization, Data curation, Formal analysis, Investigation, Methodology, Visualization, Writing – original draft, Writing – review and editing | Daryna

Kurban, Investigation, Methodology, Resources, Writing – review and editing | Simon Dufour, Conceptualization, Formal analysis, Funding acquisition, Investigation, Project administration, Supervision, Writing – review and editing.

## DATA AVAILABILITY

All 16S rRNA TAS and metagenomic sequence reads have been deposited and are available at the NCBI Sequence Read Archive under BioProject accession numbers PRJNA931348 and PRJNA931802, respectively.

## ADDITIONAL FILES

The following material is available online.

### Supplemental Material

**Supplemental Figures (mSystems00362-24-S0001.docx).** Figures S1 to S6.
**Table S1 (mSystems00362-24-s0002.csv).** Metadata for *E. coli* bovine mastitis cases.
**Table S2 (mSystems00362-24-s0003.csv).** Sample metadata for raw milk.
**Table S3 (mSystems00362-24-s0004.csv).** Sequence read data and Good's coverage for milk samples sequenced by 16S rRNA TAS.
**Table S4 (mSystems00362-24-s0005.csv).** Culture-dependent analysis of each milk sample.
**Table S5 (mSystems00362-24-s0006.csv).** Extended result of network analysis performed on 16S rRNA TAS data.
**Table S6 (mSystems00362-24-s0007.csv).** Total sequence reads produced from metagenomic sequencing.
**Table S7 (mSystems00362-24-s0008.csv).** Total sequence reads produced from metagenomic sequencing after subtracting sequence reads mapping to the bovine genome.
**Table S8 (mSystems00362-24-s0009.csv).** Sequence reads from shotgun metagenomics mapped to taxa.

### Open Peer Review

**PEER REVIEW HISTORY (review-history.pdf).** An accounting of the reviewer comments and feedback.

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
