## [Reviewer comments · mSystems]

The occurrence of *Aerococcus urinaeequi* and non-aureus Staphylococci in raw milk negatively correlates with *Escherichia coli* clinical mastitis

Dongyun Jung, Soyoun Park, Daryna Kurban, Simon Dufour, and Jennifer Ronholm

Corresponding Author(s): Jennifer Ronholm, McGill University

Review Timeline:

Submission Date:	March 11, 2024
Editorial Decision:	June 7, 2024
Revision Received:	July 22, 2024
Accepted:	August 6, 2024

Editor: Robert Beiko

Reviewer(s): Disclosure of reviewer identity is with reference to reviewer comments included in decision letter(s). The following individuals involved in review of your submission have agreed to reveal their identity: Richard Costa Polveiro (Reviewer #1)

Transaction Report:

DOI: <https://doi.org/10.1128/msystems.00362-24>

Re: mSystems00362-24 (The occurrence of *Aerococcus urinaeequi* and non-aureus *Staphylococci* in raw milk negatively correlates with *Escherichia coli* clinical mastitis)

Dear Prof. Jennifer Ronholm:

The reviewers are very supportive of the work, but had significant concerns about the reporting of methods and results. See their detailed comments attached.

Revision Guidelines

Sincerely,
Robert Beiko
Editor
mSystems

Reviewer #1 (Comments for the Author):

The article, "The occurrence of *Aerococcus urinaeequi* and non-aureus *Staphylococci* in raw milk negatively correlates with *Escherichia coli* clinical mastitis.", in Research Article format, the article addresses the relationship between the milk microbiota of healthy cows and those affected by mastitis, over certain periods, using two techniques, metataxonomy and shotgun

sequencing of specific samples. Furthermore, the study emphasizes the correlation of certain microorganisms with the state of Eubiosis and Pathobiosis of animals and their microbiota.

MAJOR COMMENTS.

1. Change the colors of the graphics so as not to harm color-blind people.
2. It is necessary to name the supplementary tables and identify the types of data they contain. Otherwise, it would be confusing and random for anyone who isn't familiar with the study and is reading it.
3. The figures have incomplete captions. Example Figure 2: Fig 2. Relative abundance of the raw milk microbiota from 26 Holstein dairy cows that developed E. coli CM. Where is the description of Figure C? The figures must come with a description of what is being displayed in that image only. The results must be presented in the Results section of the article, which must present the explanations, hypotheses and relationships between results. Describe the results in the text, in the results section, with information in the figures. Figure captions should not present a discussion of results.
4. In the methodology, explain at some point why the OTU was classified as: "Escherichia_Shigella".
5. Present more hypotheses for these bacteria found, not just as antagonists of Escherichia coli. There is not only this microorganism isolated in an infection, but others may also coexist. In a healthy clinical state, several aspects can be considered to present certain microorganisms.

MINOR COMMENTS.

INTRODUCTION

Line 62: "Based on modes of transmission, mastitis-causing organisms can be divided into two major categories: contagious or environmental..." I suggest changing to major and minor mastitis-related pathogens.

Line 108: "diversity". I suggest changing it to richness, or adding the terms broadly, with diversity and richness, as they are different terms in species ecology.

MATERIALS AND METHODS

Lines 365 - 375: "For each milk sample ... bacterial DNA extraction.". The text does not separate for the reader what would be control and what would be sick. It is difficult to understand the methodology. Are there citations for "contaminated" in two different contexts? It was not possible to understand the logic that was used. Create an organizational chart or figure better explaining the experimental design.

Line 371: "(Table S4)". It is necessary to subdivide this table. It has a lot of information in no order. Sort better to be consistent with the reader's search for information.

Lines 373-374: "Bacterial species were identified using matrix-assisted laser desorption/ionization time-of-flight (MALDI-TOF) mass spectrometry." What happened to the results of this identification technique? I couldn't find anything else about it. Why? Lines 378 -379: "Bacterial DNA was extracted from each milk sample taken from an animal that developed E. coli CM during the study period.". This information is very important. But make it clear before this point that samples were taken from animals that only had E. coli mastitis. In this subtopic, it is more important to know the details of the extractions, so that it can be reproducible by other researchers. The information on a higher milk rate for shotgun samples is very important.

Line 395: "16S rRNA gene TAS". I suggest improving this title.

Lines 445-448: "The samples selected for shotgun metagenomic sequencing included two milk samples taken from E. coli CM quarters on the day of diagnosis and 11 milk samples from healthy quarters that had more than 40% relative abundance of one of the five most abundant OTUs and low SCC (< 50,000 cells/mL)." Were the quarters from the same animal for two milk samples found positive for E. coli? Did the other animals that were considered healthy come from different herds? Different harvest times, such as time of year and food? We need more information to understand.

RESULTS

Lines 141 - 160: Much of this paragraph is methods from the article. There is a hole in the methodology that is in this paragraph of the results. Rewrite part of the methodology with this information.

Line 147: "Fig. 1." Use this figure to subdivide into two others: the first for methodology and the other for the beginning of the results. The way it is laid out makes it difficult to read.

Lines 150 -151: "Therefore, 1,127 milk samples were included in the analysis alongside kit controls (n=15) and negative PCR controls (n=15)." What are PCR controls? I don't understand, why it doesn't appear in the methodology.

Lines 168: "(Fig. S2)". I suggest following the same color pattern between the figures because they refer to the same herds between the figures.

Lines 173-174: "remained free". Like this? Free. It is being directed to the context of control animals or animals before infection. Make it more specific.

Lines 179-181: " However, when samples were analyzed without the 41 samples that produced more than 2 colonies on blood agar, Chao1 was not significantly different on the day of E. coli CM (Chao1 181 p=0.0104; Mann-Whitney statistic 236.0)." Was this result possible because the number of samples was reduced, or does it have something to do with the two aerobic colonies that were detected? Not every colony that is detected on agar must be pathogenic or contaminated, or can we say so always? This method of considering contamination or not is currently valid, but we relied on this type of approach at a time when we did not have current knowledge about microbiota. Consider rethinking that phrase.

Lines 192-195: "Other OTUs were also associated with milk from healthy quarters on the day of E. coli CM, including: Ruminococcaceae, Bifidobacterium, Bacteroides, Atopostipes, and Jeotgalicoccus (LDA > 2.0 or < -2.0; KW rank-sum test p < 0.01) (Table 1)". Consider taking these genera and species to the abstract, because they are as important as others highlighted.

Is KW an acronym? Follow the rules for presenting this before citing it.

Lines 197 - 201: "Microbial network analysis of the bacterial community". Figure S4 does not present anything as intended. I suggest putting the names of the microorganisms in the OTUs presented. Non-significance does not make the result irrelevant; on the contrary, we must consider that the sequencing may not have been perfect and the programs used for analysis as well, and this may result in non-significant data but biologically viable to be explained and hypothesized.

Lines 229: "Fig. 5 and Table S8)". Provide more information for the reader in the figure caption and text. Why the grouping on x and y axis? What do the colors in the heatmap mean?

Lines 232 - 233: "this led us to question whether certain strains of *E. coli* could act as commensal organism in the bovine udder and not result in inflammation." Perhaps this is a bit of a strong statement, because we must take into account that *E. coli* may be part of a transient microbiota, and not an indigenous one. Reconsider that sentence.

Lines 238 - 239: "The alignment showed 98.8% identity, which was above the threshold to assign OTUs, indicating that *S. marcescens* is also included in this OTU. Therefore, no evidence of a commensal mammary *E. coli* was identified in this investigation." I suggest considering that different programs and analysis methods may indicate different results. Just the comparison of OTUs does not lead to any conclusion in this case of *E. coli*, because the genomes of this genus and species are indistinguishable from other bacteria in some cases, as well as the way of assembling the MAGs genome and Metataxonomy are different, and even though Let us be conservative in this analysis, we must consider criteria to determine the similarity of 16S sequences for each case. If there is more evidence for this situation, indicate the method of analysis in the methodology.

DISCUSSION

Lines 270: "herd 2". How is Herd 2 most important and different? The text was not clear to the reader. Are there any variables different from other herds that I should consider?

Lines 299 - 300: "metagenomic sequencing". Standardize the term better throughout the text for the reader. Also taking into account "Shotgun metagenomic analysis" and "MAGs" and "MAGs".

Line 291: "UCG-005". Provide a better description of this OTU.

Line 311: "However, the inhibition of growth of *E. coli* by NAS has not been previously reported (48)." It's not clear why this quote is in this paragraph. Was the phrase about this article or the aforementioned "{48}" a statement?

Lines 341 - 342: "mechanisms that can potentially antagonize *E. coli*." The presence of these four bacteria may indicate good preventive factors in herd management; could indicate as a factor of a microbiota never before colonized by *E. coli*, and countless other possibilities. Thinking about microbiota is thinking about systems ecology. For example, reforesting a deforested area does not mean returning all native species at the same time, or otherwise, never returning native species. Recolonization has several aspects that can involve it, such as resilience. I suggest redoing this point in the conclusion and others related to it in the text. The correct study would be to force the induction of the disease and verify the resistance of this microbiota to the invading organism.

Reviewer #2 (Comments for the Author):

The authors performed a longitudinal study comparing the microbial profiles in milk samples obtained from quarters that experienced *E. coli* clinical mastitis (CM) and those without mastitis, using 16S rRNA gene sequencing and shotgun metagenomics. Overall, the manuscript is well-written, and the results are presented clearly. However, to fully understand the significance of the findings and the research performed, there are some questions (detailed below) that require further clarification.

Introduction/objectives:

The authors recently published a study to understand changes in the bacterial community in raw milk during *Staphylococcus aureus* clinical mastitis infections in dairy cattle (<https://www.ncbi.nlm.nih.gov/pmc/articles/PMC9701008/>). In the current study, the authors evaluated and compared the microbial community in milk samples obtained from quarters that experienced *E. coli* CM to those that remained mastitis-free using a similar approach and study populations as in the previously published study. Are the dairy cows included in this study also part of those that experienced *Staphylococcus aureus* clinical mastitis infections?

Line 134: "To further identify members of the microbiome negatively correlated with *E. coli* CM and to better understand the *E. coli* strains causing CM, we selected certain milk samples for analysis via shotgun metagenomics." It would be helpful for readers if the authors could clarify whether the study truly assessed the *E. coli* strains, or if the authors were referring to a species-level analysis.

Results:

Line 149-150: It would be more informative if the authors could provide additional details of the distribution of raw sequencing reads. Were these reads similar between samples obtained from quarters that developed *E. coli* CM and those that remained free from *E. coli* CM or across different time points? Please clarify the term "kit controls"; did the authors mean sampling blanks or extraction blanks, or both?

Line 152: Please clarify whether the bacteriological culture of milk samples was performed on individual quarter milk samples separately.

Line 159: It is appreciated that the authors utilized negative control samples, given the chances of contamination in milk samples due to low biomass. Did the authors mean to indicate that contaminant microbial features were identified in the negative controls? If so, were these taxa identified in the negative controls subsequently removed from the milk samples prior to analysis?

Lines 164-195: Given the observed variation from herd to herd, why not consider running multilevel linear models accounting for herd as a random effect? Additionally, since the same cows were sampled repeatedly, how was this handled in the diversity analysis? It would also be more informative if the authors could provide differences in estimates along with P-values. Which taxonomic level was used to estimate alpha and beta diversity (e.g., OTU level)? Please add this information to the text as well as in the figures.

Figure 2-Legend C is missing in the figure caption?

In the section on beta-diversity, it would be interesting to see R^2 values to understand how much variation is partitioned by each variable in the models and to conduct beta-dispersion analysis.

Figure 3-It would help readers if the authors provided the number of cows for each category and the number of samples (i.e., *E. coli* CM and healthy). Was the differential abundance analysis performed on the relative abundance of only the five most abundant OTUs?

Line 188: It is interesting that the average Shannon diversity was similar at all sampling points except on the day of *E. coli* CM. Did the authors look at other diversity metrics such as richness or evenness, and were there differences in sequence reads at these time points due to low sequencing output or a lower sample size in this category?

For the shotgun results, are the milk samples from healthy quarters also from the same time points as those from the *E. coli* CM quarters?

Line 219: It seems that the 30000729 sample yielded significantly higher sequencing reads than other samples, was that also true after the removal of bovine-associated DNA? It is not clear why all samples were not included in the MAG analysis.

Lines 248-240: Please clarify this result: "Alignment showed 98.8% identity, which was above the threshold to assign OTUs, indicating that *S. marcescens* is also included in this OTU. Therefore, no evidence of a commensal mammary *E. coli* was identified in this investigation."

Did the authors look at whether any negative control samples had OTUs assigned to the *E. coli*-*Shigella* genus in their 16S data?

Line 241: Please provide reference(s) if any.

Line 255: Figure 5 shows *E. coli* was present in only one sample (50000827). Can the authors confirm and discuss this?

Discussion:

Please use consistent terminology (microbiome or microbiota) throughout the manuscript and figures.

While the study primarily utilized 16S-based microbiome analysis, the majority of the discussion focuses on results obtained from shotgun sequencing, which was performed on a very small subset of samples. Could the authors elaborate on any potential biases or limitations and suggest further steps that could be pursued based on the results of this study to improve our understanding of the microbiome during periods of *E. coli* CM?

One of the important results found by the authors was that milk samples taken during active *E. coli* CM mastitis had significantly decreased Shannon diversity, but this quickly rebounded after the infection. Samples taken from quarters with *E. coli* CM identified a significant increase in the abundance of *Escherichia-Shigella* in milk. Also, composition largely varied from farm to farm. An important question is whether this pathogen was already present in the teat/udder of the cow and increased in relative abundance due to other factors (e.g., concurrent infection with other pathogens such as *Staphylococcus aureus* or stress), or if it is a truly new infection. Further discussion on these and how microbial changes influence *E. coli* mastitis would strengthen the manuscript.

Materials and Methods:

Line 349: While the authors provide a citation for a recently published study, did the dairy cows receive antibiotics at the beginning or during the study periods?

Line 362: Line 362: Please clarify whether milk samples were processed/cultured to identify *E. coli* in a manner similar to those with *S. aureus* (such as growth on media plates) as cited in Park et al., 2022.

Line 373: Please clarify whether *E. coli* was also identified in milk samples from healthy quarters in the MALDI results.

Line 408: It would be helpful if the authors provide information on why they opted for OTU level instead of ASV level analysis. Further, it would be interesting to see how many reads passed each step during the bioinformatics process, and the proportion of reads classified at each taxonomic level (e.g., phylum, class, genus, and species).

The authors described the use of several negative controls in the study. Back to my original questions: How were these negative controls utilized to identify potential contaminants in milk samples, especially given that milk is a low-biomass sample?

Lines 419-420: Were the alpha and beta diversity analyses conducted at the genus level or the OTU level? Since there was repeated sampling of cows and quarters, how was this addressed in the analysis?

Please clarify whether these 11 milk samples from healthy quarters are from 11 different cows.

Line 444: While the authors aimed to further analyze microbial taxa at the species level, it is unclear why only two milk samples were selected from *E. coli* CM quarters on the day of diagnosis. In a recent paper

(<https://animalmicrobiome.biomedcentral.com/articles/10.1186/s42523-022-00211-x#Sec2>), the authors mentioned n=3 milk samples for shotgun metagenomics. Are these the same samples?

Since the authors selected n=2 *E. coli* CM samples (on the day of CM) for species-level identification, this may not provide a complete picture of the dynamics of this taxon during the study period. If the authors agree, please discuss this as a potential limitation in the discussion section.

The article, “**The occurrence of *Aerococcus urinaeequi* and non-aureus *Staphylococci* in raw milk negatively correlates with *Escherichia coli* clinical mastitis.**”, in **Research Article** format, the article addresses the relationship between the milk microbiota of healthy cows and those affected by mastitis, over certain periods, using two techniques, metataxonomy and shotgun sequencing of specific samples. Furthermore, the study emphasizes the correlation of certain microorganisms with the state of Eubiosis and Pathobiosis of animals and their microbiota.

MAJOR COMMENTS.

1. Change the colors of the graphics so as not to harm color-blind people.
2. It is necessary to name the supplementary tables and identify the types of data they contain. Otherwise, it would be confusing and random for anyone who isn't familiar with the study and is reading it.
3. The figures have incomplete captions. Example Figure 2: Fig 2. Relative abundance of the raw milk microbiota from 26 Holstein dairy cows that developed *E. coli* CM. Where is the description of Figure C? The figures must come with a description of what is being displayed in that image only. The results must be presented in the Results section of the article, which must present the explanations, hypotheses and relationships between results. Describe the results in the text, in the results section, with information in the figures. Figure captions should not present a discussion of results.
4. In the methodology, explain at some point why the OTU was classified as: “*Escherichia_Shigella*”.
5. Present more hypotheses for these bacteria found, not just as antagonists of *Escherichia coli*. There is not only this microorganism isolated in an infection, but others may also coexist. In a healthy clinical state, several aspects can be considered to present certain microorganisms.

MINOR COMMENTS.

INTRODUCTION

Line 62: “Based on modes of transmission, mastitis-causing organisms can be divided into two major categories: contagious or environmental...” I suggest changing to major and minor mastitis-related pathogens.

Line 108: “diversity”. I suggest changing it to richness, or adding the terms broadly, with diversity and richness, as they are different terms in species ecology.

MATERIALS AND METHODS

Lines 365 – 375: “For each milk sample ... bacterial DNA extraction.”. The text does not separate for the reader what would be control and what would be sick. It is difficult to understand the methodology. Are there citations for “contaminated” in two different

contexts? It was not possible to understand the logic that was used. Create an organizational chart or figure better explaining the experimental design.

Line 371: "(Table S4)". It is necessary to subdivide this table. It has a lot of information in no order. Sort better to be consistent with the reader's search for information.

Lines 373-374: "Bacterial species were identified using matrix-assisted laser desorption/ionization time-of-flight (MALDI-TOF) mass spectrometry." What happened to the results of this identification technique? I couldn't find anything else about it. Why?

Lines 378 -379: "Bacterial DNA was extracted from each milk sample taken from an animal that developed E. coli CM during the study period.". This information is very important. But make it clear before this point that samples were taken from animals that only had E. coli mastitis. In this subtopic, it is more important to know the details of the extractions, so that it can be reproducible by other researchers. The information on a higher milk rate for shotgun samples is very important.

Line 395: "16S rRNA gene TAS". I suggest improving this title.

Lines 445-448: "The samples selected for shotgun metagenomic sequencing included two milk samples taken from E. coli CM quarters on the day of diagnosis and 11 milk samples from healthy quarters that had more than 40% relative abundance of one of the five most abundant OTUs and low SCC (< 50,000 cells/mL)." Were the quarters from the same animal for two milk samples found positive for E. coli? Did the other animals that were considered healthy come from different herds? Different harvest times, such as time of year and food? We need more information to understand.

RESULTS

Lines 141 – 160: Much of this paragraph is methods from the article. There is a hole in the methodology that is in this paragraph of the results. Rewrite part of the methodology with this information.

Line 147: "Fig. 1." Use this figure to subdivide into two others: the first for methodology and the other for the beginning of the results. The way it is laid out makes it difficult to read.

Lines 150 -151: "Therefore, 1,127 milk samples were included in the analysis alongside kit controls (n=15) and negative PCR controls (n=15)." What are PCR controls? I don't understand, why it doesn't appear in the methodology.

Lines 168: "(Fig. S2)". I suggest following the same color pattern between the figures because they refer to the same herds between the figures.

Lines 173-174: "remained free". Like this? Free. It is being directed to the context of control animals or animals before infection. Make it more specific.

Lines 179-181: " However, when samples were analyzed without the 41 samples that produced more than 2 colonies on blood agar, Chao1 was not significantly different on the day of E. coli CM (Chao1 181 p=0.0104; Mann-Whitney statistic 236.0)." Was this

result possible because the number of samples was reduced, or does it have something to do with the two aerobic colonies that were detected? Not every colony that is detected on agar must be pathogenic or contaminated, or can we say so always? This method of considering contamination or not is currently valid, but we relied on this type of approach at a time when we did not have current knowledge about microbiota. Consider rethinking that phrase.

Lines 192-195: “Other OTUs were also associated with milk from healthy quarters on the day of E. coli CM, including: Ruminococcaceae, Bifidobacterium, Bacteroides, Atopostipes, and Jeotgalicoccus (LDA > 2.0 or < -2.0; KW rank-sum test $p < 0.01$) (Table 1)”. Consider taking these genera and species to the abstract, because they are as important as others highlighted. Is KW an acronym? Follow the rules for presenting this before citing it.

Lines 197 – 201: “Microbial network analysis of the bacterial community”. Figure S4 does not present anything as intended. I suggest putting the names of the microorganisms in the OTUs presented. Non-significance does not make the result irrelevant; on the contrary, we must consider that the sequencing may not have been perfect and the programs used for analysis as well, and this may result in non-significant data but biologically viable to be explained and hypothesized.

Lines 229: “Fig. 5 and Table S8)”. Provide more information for the reader in the figure caption and text. Why the grouping on x and y axis? What do the colors in the heatmap mean?

Lines 232 – 233: “his led us to question whether certain strains of E. coli could act as commensal organism in the bovine udder and not result in inflammation.” Perhaps this is a bit of a strong statement, because we must take into account that E. coli may be part of a transient microbiota, and not an indigenous one. Reconsider that sentence.

Lines 238 – 239: “The alignment showed 98.8% identity, which was above the threshold to assign OTUs, indicating that S. marcescens is also included in this OTU. Therefore, no evidence of a commensal mammary E. coli was identified in this investigation.” I suggest considering that different programs and analysis methods may indicate different results. Just the comparison of OTUs does not lead to any conclusion in this case of E. coli, because the genomes of this genus and species are indistinguishable from other bacteria in some cases, as well as the way of assembling the MAGs genome and Metataxonomy are different, and even though Let us be conservative in this analysis, we must consider criteria to determine the similarity of 16S sequences for each case. If there is more evidence for this situation, indicate the method of analysis in the methodology.

DISCUSSION

Lines 270: “herd 2”. How is Herd 2 most important and different? The text was not clear to the reader. Are there any variables different from other herds that I should consider?

Lines 299 – 300: “metagenomic sequencing”. Standardize the term better throughout the text for the reader. Also taking into account “Shotgun metagenomic analysis” and “MAGs” and “MAGs”.

Line 291: “UCG-005”. Provide a better description of this OTU.

Line 311: “However, the inhibition of growth of *E. coli* by NAS has not been previously reported (48).” It's not clear why this quote is in this paragraph. Was the phrase about this article or the aforementioned “{48}” a statement?

Lines 341 – 342: “mechanisms that can potentially antagonize *E. coli*.” The presence of these four bacteria may indicate good preventive factors in herd management; could indicate as a factor of a microbiota never before colonized by *E. coli*, and countless other possibilities. Thinking about microbiota is thinking about systems ecology. For example, reforesting a deforested area does not mean returning all native species at the same time, or otherwise, never returning native species. Recolonization has several aspects that can involve it, such as resilience. I suggest redoing this point in the conclusion and others related to it in the text. The correct study would be to force the induction of the disease and verify the resistance of this microbiota to the invading organism.

Reviewer #1 (Comments for the Author):

The article, "The occurrence of Aerococcus urinaeequi and non-aureus Staphylococci in raw milk negatively correlates with Escherichia coli clinical mastitis.", in Research Article format, the article addresses the relationship between the milk microbiota of healthy cows and those affected by mastitis, over certain periods, using two techniques, metataxonomy and shotgun sequencing of specific samples. Furthermore, the study emphasizes the correlation of certain microorganisms with the state of Eubiosis and Pathobiosis of animals and their microbiota.

MAJOR COMMENTS.

1. Change the colors of the graphics so as not to harm color-blind people.

We did some research into what colours should go into Figures for colour-blind people and we re-made our figures according to these suggestions. This was a great suggestion!

2. It is necessary to name the supplementary tables and identify the types of data they contain. Otherwise, it would be confusing and random for anyone who isn't familiar with the study and is reading it.

Each of the supplementary tables have now been given a name, and a list of table names can be found in the manuscript text file as a list before the supplementary figures. The title of each supplementary table has also been added to the excel file as the first line in each tab. These names are descriptive of what data is in each table.

3. The figures have incomplete captions. Example Figure 2: Fig 2. Relative abundance of the raw milk microbiota from 26 Holstein dairy cows that developed E. coli CM. Where is the description of Figure C? The figures must come with a description of what is being displayed in that image only. The results must be presented in the Results section of the article, which must present the explanations, hypotheses and relationships between results. Describe the results in the text, in the results section, with information in the figures. Figure captions should not present a discussion of results.

The description of what was shown in Figure 2c was already present, but the (C) has now been added to the figure legend.

All Figure legends were re-written to focus on what was being displayed in the image as opposed to repeating the methods/results/or discussion sections.

4. In the methodology, explain at some point why the OTU was classified as: "Escherichia_Shigella".

The following sentence that was already in the manuscript explains this at a very broad level:

"Mothur aligned the read pairs to make contigs with the length 253 bp and the similar contigs were clustered and aligned by the SILVA reference (v. 138) (84)."

Simply, the SILVA reference database contains a taxonomy grouping at the genus level that is indicated as Escherichia-Shigella since these two organisms are indistinguishable based on 16S rRNA sequence. (Shigella is phylogenetically a sub-lineage of *E. coli* that's picked up some interesting virulence genes).

To clarify this in our revised manuscript we have added the sentence:

"Based on alignment to the SILVA reference the *E. coli* causing CM were classified as Escherichia-Shigella at the genus level."

5. Present more hypotheses for these bacteria found, not just as antagonists of Escherichia coli. There is not only this microorganism isolated in an infection, but others may also coexist. In a healthy clinical state, several aspects can be considered to present certain microorganisms.

We agree that the reason that *Aerococcus urinaeequi* and NAS are in the bovine udder microbiome is not to simply antagonize *E. coli*. However, the goal of this paper was to create a list of common bacteria in the microbiome of healthy animals, that are not commonly found in sick animals, that do not cause inflammation, and do (coincidentally) negatively correlate with *E. coli* infection. We have accomplished this goal.

Co-infections are reported, but there were not enough to analyze these independently.

Discussing the roles of *Aerococcus* and NAS in the ecology of the udder microbiome is far beyond the scope of this paper.

MINOR COMMENTS.

INTRODUCTION

Line 62: "Based on modes of transmission, mastitis-causing organisms can be divided into two major categories: contagious or environmental..." I suggest changing to major and minor mastitis-related pathogens.

We chose to leave this sentence as is. Our reasoning is that in all existing literature on mastitis the pathogenic organisms are classified as contagious or environmental. This is

common nomenclature in the mastitis literature. It would add unnecessary complexity to the literature to change the wording in just this paper.

Line 108: "diversity". I suggest changing it to richness, or adding the terms broadly, with diversity and richness, as they are different terms in species ecology.

We have changed the word "diversity" to "richness" as suggested.

MATERIALS AND METHODS

Lines 365 - 375: "For each milk sample ... bacterial DNA extraction.". The text does not separate for the reader what would be control and what would be sick. It is difficult to understand the methodology. Are there citations for "contaminated" in two different contexts? It was not possible to understand the logic that was used. Create an organizational chart or figure better explaining the experimental design.

We have removed all references to samples from *E. coli* mastitis quarters as contaminated.

The word contaminated used strictly to refer to any samples which produced >2 colony morphologies on a blood agar plate.

We have improved the clarity of the text for lines 365-375:

"The study was affiliated with Park *et al.*, as part of a larger project that included 698 Holstein dairy cows from five herds in the Quebec province, in proximity to the Faculty of Veterinary Medicine at Université de Montréal in Saint-Hyacinthe (56). Raw milk samples were collected aseptically from each quarter every two weeks during the lactation period between December 2018 and March 2020. Briefly, the farm's teat disinfectant was applied, after a contact time of 20 seconds, the teats were dried using paper towel and each teat end was then scrubbed with individual alcohol swabs. Approximately 60 mL of milk was then discarded, and 50 mL were collected (i.e., cisternal milk). Milk samples were placed on ice and transported, on the same day, to the Faculty of Veterinary Medicine laboratory where they were stored at 4°C prior to being aliquoted for the different parts of the study. After being aliquoted samples were frozen. The SCC was measured for each milk sample from quarters without any clinical signs of mastitis. In parallel to these farm visits, CM was identified by the producers by visual inspection of the milk and udders, and a milk sample was collected from the affected quarter by the producers at diagnosis, following the protocol previously described. These latter samples were immediately frozen on the farm at -20°C, collected by the research team during the next farm visit, and processed in the laboratory as previously described.

The National Mastitis Council's guidance on raw milk sampling advises that if a milk sample is plated on blood agar, and after 24 or 48 hours (when the sample yielded no growth after 24h of incubation) of incubation >2 different colony morphologies are

observed, the sample is considered contaminated (55). Therefore, after collection, 10 μ L of each raw milk sample was spread on 5% sheep blood agar and incubated for 24 or 48 hours at 35°C. The number of phenotypically different colonies that were grown for each milk sample was counted and recorded (Table S4) (81). If a sample produced >2 colony morphologies, colony identification was not performed on these samples, and they were classified as contaminated (Figure S6). Colony identification was not performed on samples which produced >2 colony morphologies. However, DNA for 16S rRNA TAS was still extracted from these samples and data set was analyzed with and without putatively contaminated samples. For samples which produced 2 or less colony morphologies on blood agar, a representative colony for each colony type was identified using matrix-assisted laser desorption/ionization time-of-flight (MALDI-TOF) mass spectrometry (Table S4). All the collected milk samples were then stored between -10°C and -20°C until bacterial DNA extraction.

Sample were classified as being from a healthy quarter if they were collected from a quarter with no clinical symptoms of mastitis and a SCC of <200,000 cells/mL of milk. If a quarter milk sample came from a quarter which had signs of CM, the sample was classified as 'clinical mastitis. If a quarter milk sample had a SCC of >200,000 cells/mL of milk and yielded a positive culture for *E. coli* the sample was classified as having subclinical mastitis caused by *E. coli* (Figure S6)."

We have also added a supplementary figure (Figure S6) to indicate how samples were classified as sick, healthy, or contaminated:

Line 371: "(Table S4)". It is necessary to subdivide this table. It has a lot of information in no order. Sort better to be consistent with the reader's search for information.

We feel that keeping Table S4 as it is important to the overall story of the paper. Having the number of colony morphologies observed alongside colony identification is a very good way to communicate this data. It flows – here are how many colonies we counted and then here's what they were identified as.

However, we have changed the table and manuscript to better explain this table to the reader. The following changes were made:

The table heading was changed to "Number of Colony Morphologies Observed on Blood Agar" to make it clear that this number indicates how many different types of colonies were observed on the plate. Another heading was changed to "Interpretation of Colony Morphology Number" to make it clear how we interpreted the number of colony morphologies on blood agar.

With the changes from the previous comment, we think this is now clear.

Lines 373-374: "Bacterial species were identified using matrix-assisted laser desorption/ionization time-of-flight (MALDI-TOF) mass spectrometry." What happened to the results of this identification technique? I couldn't find anything else about it. Why?

This was used to confirm *E. coli* mastitis, any co-infections, and identify each of the colonies from blood agar plates with 2 or less colonies. The identification of each of the different colonies is shown with the colony counts from each plate in Table S4.

This is now indicated by the following sentence:

"For samples which produced 2 or less colony morphologies on blood agar, a representative colony for each colony type was identified using matrix-assisted laser desorption/ionization time-of-flight (MALDI-TOF) mass spectrometry (Table S4)."

Lines 378 -379: "Bacterial DNA was extracted from each milk sample taken from an animal that developed *E. coli* CM during the study period.". This information is very important. But make it clear before this point that samples were taken from animals that only had *E. coli* mastitis. In this subtopic, it is more important to know the details of the extractions, so that it can be reproducible by other researchers. The information on a higher milk rate for shotgun samples is very important.

Samples were taken from 698 animals – most of which did not have *E. coli* mastitis at any point during the study. Only 10 animals of this 698 had *E. coli* mastitis. Bacterial DNA was only extracted from milk taken from animals that developed *E. coli* mastitis at some point in the study, as stated:

"Bacterial DNA was extracted from each milk sample taken from an animal that developed *E. coli* CM during the study period."

To clarify the shotgun metagenomics part, we added the following sentences:

"To extract a higher quantity of DNA for shotgun metagenomic sequencing, a 6.0 mL aliquot of milk was used. The sample was prepared by centrifuging 1 mL of milk at 16,000xg for 10 minutes, discarding the supernatant, then adding another 1 mL of fresh milk to the pellet and repeating this procedure five times. The DNA was extracted from the larger pellet using the same protocols as described above."

Line 395: "16S rRNA gene TAS". I suggest improving this title.

We changed the title to 16S rRNA gene sequencing.

Lines 445-448: "The samples selected for shotgun metagenomic sequencing included two milk samples taken from E. coli CM quarters on the day of diagnosis and 11 milk samples from healthy quarters that had more than 40% relative abundance of one of the five most abundant OTUs and low SCC (< 50,000 cells/mL)." Were the quarters from the same animal for two milk samples found positive for E. coli? Did the other animals that were considered healthy come from different herds? Different harvest times, such as time of year and food? We need more information to understand.

The samples used for metagenomics are indicated in Figure 1. From this figure you can understand which animals the samples are from, overlap in samples between healthy cows and cows with bovine mastitis, and differences in collection period. To make this clear, we have added the sentence:

"Samples used for metagenomic sequences are indicated in Figure 1."

RESULTS

Lines 141 - 160: Much of this paragraph is methods from the article. There is a hole in the methodology that is in this paragraph of the results. Rewrite part of the methodology with this information.

We disagree. The results from the culturing work and the quality metrics of our sequencing work belong in the results section. Then we explain how we used these results to decide which samples to include in the analysis – this wouldn't make sense in the materials and methods section. This is especially true in a journal format where the results come before the M&M section. The details of the actual analysis are in the M&M Section.

We did move a few sections of sentences of this paragraph to the M&M section based on other comments from you.

Line 147: "Fig. 1." Use this figure to subdivide into two others: the first for methodology and the other for the beginning of the results. The way it is laid out makes it difficult to read.

We disagree. This figure indicates the results of our culturing and diagnostic work and is a clear way to understand where the samples came from. We don't see how Figure 1 could be split without losing the ability to visualize the source of and analysis applied to >1000 samples in a single glance.

Lines 150 -151: "Therefore, 1,127 milk samples were included in the analysis alongside kit controls (n=15) and negative PCR controls (n=15)." What are PCR controls? I don't understand, why it doesn't appear in the methodology.

This sentence was moved to the M&M section and was expanded to the following sentence to explain:

"An independent negative PCR control (n=15), which consisted of an attempt to amplify DNA/RNA free water, was included for each 96-well PCR reaction performed as part of this study. Both kit controls and PCR controls were also sequenced."

Lines 168: "(Fig. S2)". I suggest following the same color pattern between the figures because they refer to the same herds between the figures.

There are no other figures that specifically compare the composition of the microbiome between herds.

Lines 173-174: "remained free". Like this? Free. It is being directed to the context of control animals or animals before infection. Make it more specific.

This sentence was changed to:

"The diversity of the milk microbiome was compared over time between quarters which developed *E. coli* CM and control quarters that did not develop *E. coli* CM during the study period."

Lines 179-181: " However, when samples were analyzed without the 41 samples that produced more than 2 colonies on blood agar, Chao1 was not significantly different on the day of *E. coli* CM (Chao1 181 p=0.0104; Mann-Whitney statistic 236.0)." Was this result possible because the number of samples was reduced, or does it have something to do with the two aerobic colonies that were detected? Not every colony that is detected on agar must be pathogenic or contaminated, or can we say so always? This method of considering contamination or not is currently valid, but we relied on this type of approach at a time when we did not have current knowledge about microbiota. Consider rethinking that phrase.

This point is well taken. Yes, we lose statistical significance because we reduced the number of samples. We agree that not every sample with ≥ 3 colony morphologies detected on blood agar is contaminated and have addressed this issue in our own research and are almost ready to publish another paper agreeing with your comment.

This sentence doesn't quite reflect as antiquated view of the microbiome as you may think at first glance. The healthy milk microbiome is at low enough biomass / concentration you would not expect to observe >2 colonies on a blood agar plate inoculated with 10uL of milk in a not contaminated sample. In most healthy milk samples, you don't see any colonies on blood agar. However, if fecal material (which is VERY common in the environment where the samples are collected) enters the milk – the bacteria are at a much higher concentration and easily detectable on blood agar.

To meet the standards of the milk microbiome community (who also commonly read and review our papers) it is imperative that we follow the rules of the "Laboratory Handbook on Bovine Mastitis (LHBM)" This handbook requires that all samples with more than 2 colony morphologies be discarded – we have had papers rejected for not following this rule. **The reality is that most samples with >2 colony morphologies are probably contaminated.** But we accept the possibility that you could have a high-density milk microbiome (without contamination). Our compromise is to simply analyze the samples with and without the samples that are "contaminated" by LHBM standards and report where there is a difference. There is very rarely a difference. In this paper this small decrease in statistical significance of the Chao1 was the only difference we observed when we analyzed sequencing data with and without the "contaminated" samples.

Lines 192-195: "Other OTUs were also associated with milk from healthy quarters on the day of E. coli CM, including: Ruminococcaceae, Bifidobacterium, Bacteroides, Atopostipes, and Jeotgalicoccus (LDA > 2.0 or < -2.0; KW rank-sum test p < 0.01) (Table 1)". Consider taking these genera and species to the abstract, because they are as important as others highlighted. Is KW an acronym? Follow the rules for presenting this before citing it.

Good suggestion, but they were not as significant as the others, and we don't want to over present our data. But, we have decided to follow up on these taxa in our follow up studies.

We have corrected KW to "Kruskal-Wallis".

Lines 197 - 201: "Microbial network analysis of the bacterial community". Figure S4 does not present anything as intended. I suggest putting the names of the microorganisms in the OTUs presented. Non-significance does not make the result irrelevant; on the contrary, we must consider that the sequencing may not have been perfect and the programs used for analysis as well, and this may result in non-significant data but biologically viable to be explained and hypothesized.

We agree that non-significance is important – thus why this information is included in the manuscript. Unfortunately, there is no way to put the names of taxa in the figure and make

it look presentable. Table S5 has the names of each of the OTUs in the figure and this is indicated in the text. It is easily accessible to anyone who is interested.

We have added the following sentence to the figure legend of Figure S4:

"The phylogenetic identity at the genus level, of each of the OTUs identified in this Figure are listed in Table S5."

Lines 229: "Fig. 5 and Table S8)". Provide more information for the reader in the figure caption and text. Why the grouping on x and y axis? What do the colors in the heatmap mean?

We have changed the figure legend for Figure 5 to:

"Fig 5. Shotgun metagenomic analysis of the microbial composition of healthy and *E. coli* CM samples at species-level. The relative abundance of taxonomic composition of the metagenomic sequencing is shown using a heat map where sample IDs are displayed on the x-axis and bacterial species with > 0.1% of the relative abundance of sequence reads are shown on the y-axis. The phylogenetic relatedness of the bacterial species identified is shown on the left y-axis, and the phylogenetic relatedness of the metagenomic sequences is shown at the top of the figure."

The legend for the colours is shown at the top and indicates the % relative abundance of sequence reads that mapped to a particular species. As an example, *E. coli* is shown in white for samples 41000070 and 51000029 since 100% of the sequence reads in each of these samples mapped to *E. coli*.

Lines 232 - 233: "his led us to question whether certain strains of E. coli could act as commensal organism in the bovine udder and not result in inflammation." Perhaps this is a bit of a strong statement, because we must take into account that E. coli may be part of a transient microbiota, and not an indigenous one. Reconsider that sentence.

We agree with your statement. However, if you continue reading the paragraph goes on to explain that this sample did not have a high abundance of *E. coli* but rather a high abundance of *S. marcescens*. Thus, even though we looked extensively we were unable to find a milk sample with a low SCC and a high abundance of *E. coli*, providing evidence that *E. coli* is probably always inflammatory in the bovine udder.

Lines 238 - 239: "The alignment showed 98.8% identity, which was above the threshold to assign OTUs, indicating that S. marcescens is also included in this OTU. Therefore, no evidence of a commensal mammary E. coli was identified in this

investigation." I suggest considering that different programs and analysis methods may indicate different results. Just the comparison of OTUs does not lead to any conclusion in this case of *E. coli*, because the genomes of this genus and species are indistinguishable from other bacteria in some cases, as well as the way of assembling the MAGs genome and Metataxonomy are different, and even though Let us be conservative in this analysis, we must consider criteria to determine the similarity of 16S sequences for each case. If there is more evidence for this situation, indicate the method of analysis in the methodology.

Yes- different programs may yield different results.

We agree that the comparison of OTUs does not lead to a conclusion. However, assembling a MAG of *S. marcescens* where almost 100% of the sequence reads map to *S. marcescens* (Figure 5) from a sample which had almost 100% of an OTU assigned to *E. coli* is reasonably good evidence that there was a high occurrence of *S. marcescens* in a sample that was mistakenly grouped with *E. coli*. This is especially true since our 16S rRNA region between the two organisms is indistinguishable based on the parameters we used. It's three lines of evidence that we understand what's going on here.

DISCUSSION

Lines 270: "herd 2". How is Herd 2 most important and different? The text was not clear to the reader. Are there any variables different from other herds that I should consider?

Herd 2 just had more Actinobacteria than the other herds – which is what we said (Figure 2), it's not most important. We have some guesses about why based on their bedding, but herd 2 was the only farm that used their bedding type so nothing is significant and we are hesitant to put this in the text because it is just a guess on our part.

Lines 299 - 300: "metagenomic sequencing". Standardize the term better throughout the text for the reader. Also taking into account "Shotgun metagenomic analysis" and "MAGs" and "MAGs".

We removed the word "shotgun" and just went with "metagenomic sequencing" and "metagenomic sequencing analysis."

MAGs refer specifically to when we tried to assemble genomes from the metagenomic sequence reads. Metagenomic sequencing analysis also refers to other types of metagenomic sequencing analyses.

Line 291: "UCG-005". Provide a better description of this OTU.

It is likely an uncultured Oscillospiraceae.

In line 188 we already have indicated that it's from the Oscillospiraceae. We do not know anything beyond this for this OTU.

Line 311: "However, the inhibition of growth of *E. coli* by NAS has not been previously reported (48)." It's not clear why this quote is in this paragraph. Was the phrase about this article or the aforementioned "{48}" a statement?

The paragraph indicates that other studies (such as reference 48) have shown that the presence of NAS can inhibit *S. aureus* growth and have a positive impact in preventing *S. aureus* mastitis. However, our current article is the first to show that the presence of NAS may also have a positive impact in preventing *E. coli*.

To make this clearer we have removed reference 48 – because it isn't the only reference for NAS, and revised the sentence as follows:

"However, the inhibition of growth of *E. coli* by NAS has not been previously reported prior to this study."

Lines 341 - 342: "mechanisms that can potentially antagonize *E. coli*." The presence of these four bacteria may indicate good preventive factors in herd management; could indicate as a factor of a microbiota never before colonized by *E. coli*, and countless other possibilities. Thinking about microbiota is thinking about systems ecology. For example, reforesting a deforested area does not mean returning all native species at the same time, or otherwise, never returning native species. Recolonization has several aspects that can involve it, such as resilience. I suggest redoing this point in the conclusion and others related to it in the text. The correct study would be to force the induction of the disease and verify the resistance of this microbiota to the invading organism.

This point is well taken, and we are currently carrying out studies to build a "resilient" microbiome as defined by the current work and force the introduction of the disease as you suggest.

However, while the presence of these 4 bacteria may well indicate good preventative factors in herd management – there were no herds in our study which did not have a case of *E. coli* mastitis. And indeed the *E. coli* mastitis quarters and the control quarters were on the same cow – thus examining (and finding) potential mechanisms in our differentially present bacteria is not unreasonable.

Reviewer #2 (Comments for the Author):

The authors performed a longitudinal study comparing the microbial profiles in milk samples obtained from quarters that experienced *E. coli* clinical mastitis (CM) and those without mastitis, using 16S rRNA gene sequencing and shotgun metagenomics. Overall, the manuscript is well-written, and the results are presented clearly. However, to fully understand the significance of the findings and the research performed, there are some questions (detailed below) that require further clarification.

Introduction/objectives:

The authors recently published a study to understand changes in the bacterial community in raw milk during *Staphylococcus aureus* clinical mastitis infections in dairy cattle (<https://www.ncbi.nlm.nih.gov/pmc/articles/PMC9701008/>). In the current study, the authors evaluated and compared the microbial community in milk samples obtained from quarters that experienced *E. coli* CM to those that remained mastitis-free using a similar approach and study populations as in the previously published study. Are the dairy cows included in this study also part of those that experienced *Staphylococcus aureus* clinical mastitis infections?

Yes. It's the same sampling campaign.

However, we sampled from 698 cows, and only analyzed the milk from cows that had *S. aureus* infections (n=10) in the preceding paper and only the cows that had *E. coli* in the current paper (n=26) (and *K pneumoniae* in our upcoming paper).

Only one cow had both *S. aureus* and *E. coli* infections (H4C88) and was therefore included in both studies. Figure 1 in this paper can be compared to Figure S1 in the previous paper to understand the limited overlap in samples between the two studies.

Line 134: "To further identify members of the microbiome negatively correlated with *E. coli* CM and to better understand the *E. coli* strains causing CM, we selected certain milk samples for analysis via shotgun metagenomics." It would be helpful for readers if the authors could clarify whether the study truly assessed the *E. coli* strains, or if the authors were referring to a species-level analysis.

We removed the word "strain" from this sentence, since we cannot defend that we are truly assessing strains instead of a species-level analysis.

Results:

Line 149-150: It would be more informative if the authors could provide additional details of the distribution of raw sequencing reads. Were these reads similar between samples obtained from quarters that developed *E. coli* CM and those that remained

free from E. coli CM or across different time points? Please clarify the term "kit controls"; did the authors mean sampling blanks or extraction blanks, or both?

We didn't include the distribution of raw sequencing reads since we rarified to 3,100 reads per sample before analysis – which kind of negates knowing how many reads were dedicated to each sample to start.

I'm not sure what a sampling blank is.

We clarify kit controls in this revised sentence:

"A negative kit control (n=15), which included extracting DNA from DNA/RNA free water using each of the reagents available in the DNA extraction kit, was prepared for each DNA extraction kit used. An independent negative PCR control (n=15), which consisted of an attempt to amplify DNA/RNA free water, was included for each 96-well PCR reaction performed as part of this study. Both kit controls and PCR controls were also sequenced."

Line 152: Please clarify whether the bacteriological culture of milk samples was performed on individual quarter milk samples separately.

Each individual quarter sample was plated separately.

We have revised the sentence to:

"In addition, each individual quarter sample was plated separately on blood agar, 41/1,127 samples produced more than 2 different colony morphologies."

Line 159: It is appreciated that the authors utilized negative control samples, given the chances of contamination in milk samples due to low biomass. Did the authors mean to indicate that contaminant microbial features were identified in the negative controls? If so, were these taxa identified in the negative controls subsequently removed from the milk samples prior to analysis?

No evidence of contamination was seen. We have clarified this in the text:

"No evidence of microbial or nucleic acid contamination was observed in any of the reagents used for DNA extraction or PCR amplification based on the sequencing results for several negative controls (Fig. S1)."

This question is addressed further in your final question about the negative controls.

Lines 164-195: Given the observed variation from herd to herd, why not consider

running multilevel linear models accounting for herd as a random effect? Additionally, since the same cows were sampled repeatedly, how was this handled in the diversity analysis? It would also be more informative if the authors could provide differences in estimates along with P-values. Which taxonomic level was used to estimate alpha and beta diversity (e.g., OTU level)? Please add this information to the text as well as in the figures.

OTU was used to estimate alpha and beta diversity. This has now been made clear through the text and in the figure legends.

Our diversity analysis considered each sample as independent.

The goal of this study was to compare the microbiome between quarters that were infected and not infected across time. We think what we have done is the most accurate way possible.

We did analyze the data set multiple ways using multiple different software and the same basic conclusions were reached each time. We also want this paper to be somewhat comparable to our *S. aureus* paper which you cite many times.

Figure 2-Legend C is missing in the figure caption?

The following figure caption has been added:

“(C) The relative abundance of each of the five most common OTUs is shown for each of the 5 herds included in the study.”

In the section on beta-diversity, it would be interesting to see R² values to understand how much variation is partitioned by each variable in the models and to conduct beta-dispersion analysis.

We added the R² values to the figure legend since they made the figure look crowded if we added them using big enough text to read.

Figure 3-It would help readers if the authors provided the number of cows for each category and the number of samples (i.e., *E. coli* CM and healthy). Was the differential abundance analysis performed on the relative abundance of only the five most abundant OTUs?

The number of cows and therefore the number of healthy and *E. coli* CM samples varies based on what part of Figure 3 you're looking at. In Figure 3a you can see the number of

cows included. For all *E. coli* CM samples, the number of cows = 23. One quarter is infected and the other 3 are “healthy” in each instance.

Differential abundance analysis was performed using all OTUs. We have clarified this with the following sentence in the figure legend for Figure 3:

“LDA score (D) for the five most common OTUs in the study to demonstrate differences between milk samples taken from cows with and without *E. coli* CM – all OTUs in the study were considered in the differential abundance analysis.”

Line 188: It is interesting that the average Shannon diversity was similar at all sampling points except on the day of *E. coli* CM. Did the authors look at other diversity metrics such as richness or evenness, and were there differences in sequence reads at these time points due to low sequencing output or a lower sample size in this category?

Yes. We also used Chao1 to examine richness. This is reported in lines 187-192. Chao1 was statistically significantly different on the day of *E. coli* CM – but not on any of the other days. However, we lost this significance if we removed the 41 samples that were “contaminated” based on the standards of the National Mastitis Council.

For the shotgun results, are the milk samples from healthy quarters also from the same time points as those from the *E. coli* CM quarters?

No. The samples chosen for shot gun metagenomics were based on the abundance of specific taxa that we wanted to assemble different MAGs for. We have indicated the samples used for shot gun metagenomics in Figure 1 using triangles to make it easy to see where these samples were pulled from relative to the entire sample set.

Line 219: It seems that the 30000729 sample yielded significantly higher sequencing reads than other samples, was that also true after the removal of bovine-associated DNA? It is not clear why all samples were not included in the MAG analysis.

Yes. The numbers reported in Table S7 are after bovine DNA was removed. We did include all of the samples in MAG analysis, but we only reported samples from which we successfully obtained MAGs. We do not know why this sample was so successfully sequenced relative to the other samples; they were all processed in the same way.

We added the following sentence to clarify:

“All samples underwent MAG analysis, however, only sample, 30000729, yielded high-quality MAGs.”

Lines 248-240: Please clarify this result: "Alignment showed 98.8% identity, which was above the threshold to assign OTUs, indicating that *S. marcescens* is also included in this OTU. Therefore, no evidence of a commensal mammary *E. coli* was identified in this investigation."

We added several sentences to clarify our thinking in getting to this sentence. Hopefully, this answers your question:

"Milk sample 50000827 was selected for metagenomic sequencing since it was identified as having a high abundance of the *Escherichia-Shigella* OTU (90.42%), but a low SCC/ mL of milk of only 44,000. The combination of a high abundance of *E. coli* in milk combined with a low SCC led us to question if certain strains of *E. coli* could act as commensal organism in the bovine udder and not result in inflammation – something that has not been reported. There were several milk samples in our dataset that had a high relative abundance of *Escherichia-Shigella* but low SSC (Fig. 4), but we chose to perform metagenomic sequencing 50000827 because the extremely high relative abundance of the *Escherichia-Shigella* OTU in the sample would allow for the best chance at assembling a MAG. Metagenomic sequencing revealed a single MAG of *S. marcescens* (Genome size: 2,840,286 bp, CDS: 2,672; completeness = 71.29%, contamination = 1.84%) from this sample, no *E. coli* was identified from the metagenomic sequences. This led us to question if there was enough similarity between the V4 region of *Escherichia-Shigella* and *S. marcescens* that *S. marcescens* 16S rRNA sequences would be assigned to this OTU. To examine the similarity of V4 region in 16S rRNA gene between *E. coli* and *S. marcescens*, we aligned the sequences of the V4 region (Fig. S5). The alignment showed 98.8% identity, which was above the threshold to assign OTUs, indicating that *S. marcescens* sequences are also included in the *Escherichia-Shigella* OTU. Therefore, no evidence of a commensal mammary *E. coli* was identified in this investigation."

Did the authors look at whether any negative control samples had OTUs assigned to the *E. coli*-*Shigella* genus in their 16S data?

Very few of the negative control samples had small numbers of *E. coli*-*Shigella* genus (OTU3) assigned to them, most had 0 OTUs assigned to this genus. We have shown you the raw data with this highlighted in response to another comment a little further down.

The negative control with the most sequences mapping to this OTU had 381 reads, while most milk samples from cows with *E. coli* mastitis had >10,000 reads assigned to this genus (before rarefaction).

Line 241: Please provide reference(s) if any.

There is no reference for this. We are simply reporting what is in our MAGs.

We explain why we pay attention to these genes in the discussion with references in the following sentences:

"The MAGs of *S. auricularis*, *S. haemolyticus*, and *A. urinaeequi* were constructed from a healthy milk sample, and each genome encodes D- and L-lactate dehydrogenase which catalyzes the production of lactic acid from pyruvate. Several studies have reported that lactic acid can inhibit the growth of *E. coli* O157:H7 *in vitro* (70–72)."

Line 255: Figure 5 shows E. coli was present in only one sample (50000827). Can the authors confirm and discuss this?

Figure 5 shows *E. coli* was present in samples 41000070 and 51000029 – as expected; and that sample 50000827 mostly contained *S. marcescens* and *P. fluorescens* to a lesser extent (as explained in the text).

We have changed the colour scheme and revised the figure legend to make this figure clearer.

Discussion:

Please use consistent terminology (microbiome or microbiota) throughout the manuscript and figures.

We have replaced the word microbiota with microbiome.

While the study primarily utilized 16S-based microbiome analysis, the majority of the discussion focuses on results obtained from shotgun sequencing, which was performed on a very small subset of samples. Could the authors elaborate on any potential biases or limitations and suggest further steps that could be pursued based on the results of this study to improve our understanding of the microbiome during periods of E. coli CM?

A major limitation of our approach was the lack of culture-based work to follow up on the hypothesis about inter-bacterial interactions generated by this study. Therefore, we have added the following sentence:

"A major limitation of the current study is that co-culture tests to determine if antagonistic activity does exist between *Staphylococcus* and *Aerococcus* isolates and *E. coli* in the context of the bovine udder were not performed. However, this will be the focus of much future work."

One of the important results found by the authors was that milk samples taken during active *E. coli* CM mastitis had significantly decreased Shannon diversity, but this quickly rebounded after the infection. Samples taken from quarters with *E. coli* CM identified a significant increase in the abundance of *Escherichia-Shigella* in milk. Also, composition largely varied from farm to farm. An important question is whether this pathogen was already present in the teat/udder of the cow and increased in relative abundance due to other factors (e.g., concurrent infection with other pathogens such as *Staphylococcus aureus* or stress), or if it is a truly new infection. Further discussion on these and how microbial changes influence *E. coli* mastitis would strengthen the manuscript.

We are hesitant to comment on this since our data set doesn't include stress indicators. In addition, we cannot achieve species level identification with 16S rRNA sequencing, so there is no way to know if *S. aureus* (or other pathogens identified at the species level) are in the milk in the weeks prior to *E. coli*. As soon as *E. coli* counts were high enough to be identified by culture-dependent work it coincided with increases in inflammation and the cows were classified as having mastitis.

Materials and Methods:

Line 349: While the authors provide a citation for a recently published study, did the dairy cows receive antibiotics at the beginning or during the study periods?

Cows did receive antibiotics during the study period – but for the 698 cows in the study, there was almost 698 different treatments for antibiotics. Any infections were treated, herds 1, 2 & 4 also did blanket dry cow therapy, herds 3 & 5 did selective treatment at dry-off. Combinations of Cefa-Dri and Novov dry were used. There is no pattern consistent enough to analyze or report. This was the same for Park et al., 2022 – thus why nothing is reported in that paper either.

Line 362: Line 362: Please clarify whether milk samples were processed/cultured to identify *E. coli* in a manner similar to those with *S. aureus* (such as growth on media plates) as cited in Park et al., 2022.

The samples were processed identically to that in Park et al., 2022. We have revised the entire text through this section to make it clearer.

Line 373: Please clarify whether *E. coli* was also identified in milk samples from healthy quarters in the MALDI results.

E. coli was not identified from milk samples that were classified as "healthy" based on the decision chart included in our new Figure S6 (suggested by the other reviewer).

Line 408: It would be helpful if the authors provide information on why they opted for OTU level instead of ASV level analysis. Further, it would be interesting to see how many reads passed each step during the bioinformatics process, and the proportion of reads classified at each taxonomic level (e.g., phylum, class, genus, and species).

We opted for OTUs to remain consistent with Park et al., 2022 and O'Brien et al., (the sister paper on *Klebsiella* which is soon to be submitted). Although ASVs are becoming more popular OTUs are still a valid way to analyze this type of data. We are not sure what value adding how many reads passed each step would bring to the paper. We feel that it would further complicate an already very long and already very complicated paper.

The authors described the use of several negative controls in the study. Back to my original questions: How were these negative controls utilized to identify potential contaminants in milk samples, especially given that milk is a low-biomass sample?

Contamination in milk samples was controlled using the approved blood agar growth test that is described in length in the manuscript and in this review.

Contamination in the extraction kit and PCR reagents were controlled by sequencing a negative control for both the extraction reactions for each kit used and a negative PCR reaction for each plate amplified. There was no evidence of contamination in either.

Please find below screenshots of our OTU count table for some milk samples and for the 21 most common OTUs. Compare this to the second screenshot of the OTUs for the negative controls. Most of the negatives have no sequences for most OTUs identified in our milk samples.

We did not remove the ones that did based on the very low abundance in the negative control sequences based on this advice:

<https://mothur.org/blog/2014/TheKitome/>

The alpha and beta diversity analyses were conducted at the OTU level. Each sample was treated as independent.

Please clarify whether these 11 milk samples from healthy quarters are from 11 different cows.

These samples are taken from 11 different cows. This is displayed in Figure 1.

We have revised the sentence as follows:

"The samples selected for metagenomic sequencing included two milk samples taken from E. coli CM quarters on the day of diagnosis, and 11 milk samples from healthy quarters each from different cows."

Line 444: While the authors aimed to further analyze microbial taxa at the species level, it is unclear why only two milk samples were selected from E. coli CM quarters on the day of diagnosis. In a recent paper (<https://animalmicrobiome.biomedcentral.com/articles/10.1186/s42523-022-00211-x#Sec2>), the authors mentioned n=3 milk samples for shotgun metagenomics. Are these the same samples?

These are not the same samples. We selected each of our metagenomic sequencing samples for different reasons to answer different questions.

In Park et al., we were trying to get MAGs to identify potential *S. aureus* antagonists at the species level. In the current paper we wanted to try to assemble MAGs of pathogenic and non-pathogenic *E. coli* based on SCC – and assemble MAGs of potential *E. coli* antagonists. Ultimately, we were unable to assemble *E. coli* MAGs and found that the potential non-pathogenic *E. coli* was actually identified as *S. marcescens*, but we did identify some potential antagonists.

Since the authors selected n=2 E. coli CM samples (on the day of CM) for species-level identification, this may not provide a complete picture of the dynamics of this taxon during the study period. If the authors agree, please discuss this as a potential limitation in the discussion section.

We absolutely agree! We are addressing the dynamics of *E. coli* populations in mastitis in a future manuscript. It too complicated to do justice for in this paper. Also, based on the data in this paper anything we say about having multiple lineages of *E. coli* in a co-infection as being the reason we couldn't assemble MAGs would just be a guess. We do not have enough data to discuss this in this paper.

Re: mSystems00362-24R1 (The occurrence of *Aerococcus urinaeequi* and non-aureus *Staphylococci* in raw milk negatively correlates with *Escherichia coli* clinical mastitis)

Dear Prof. Jennifer Ronholm:

Your manuscript has been accepted, and I am forwarding it to the ASM production staff for publication. Your paper will first be checked to make sure all elements meet the technical requirements. ASM staff will contact you if anything needs to be revised before copyediting and production can begin. Otherwise, you will be notified when your proofs are ready to be viewed.

Sincerely,
Robert Beiko